# Testing Closeness With Unequal Sized Samples

**Bhaswar B. Bhattacharya**
Department of Statistics
Stanford University
California, CA 94305
bhaswar@stanford.edu

**Gregory Valiant**[*]
Department of Computer Science
Stanford University
California, CA 94305
valiant@stanford.edu

## Abstract

We consider the problem of testing whether two unequal-sized samples were drawn from identical distributions, versus distributions that differ significantly. Specifically, given a target error parameter $\varepsilon > 0$, $m_1$ independent draws from an unknown distribution $p$ with discrete support, and $m_2$ draws from an unknown distribution $q$ of discrete support, we describe a test for distinguishing the case that $p = q$ from the case that $||p - q||_1 \geq \varepsilon$. If $p$ and $q$ are supported on at most $n$ elements, then our test is successful with high probability provided $m_1 \geq n^{2/3}/\varepsilon^{4/3}$ and $m_2 = \Omega\left(\max\{\frac{n}{\sqrt{m_1}\varepsilon^2}, \frac{\sqrt{n}}{\varepsilon^2}\}\right)$. We show that this tradeoff is information theoretically optimal throughout this range in the dependencies on all parameters, $n, m_1$, and $\varepsilon$, to constant factors for worst-case distributions. As a consequence, we obtain an algorithm for estimating the mixing time of a Markov chain on $n$ states up to a $\log n$ factor that uses $\tilde{O}(n^{3/2}\tau_{mix})$ queries to a "next node" oracle. The core of our testing algorithm is a relatively simple statistic that seems to perform well in practice, both on synthetic and on natural language data. We believe that this statistic might prove to be a useful primitive within larger machine learning and natural language processing systems.

## 1  Introduction

One of the most basic problems in statistical hypothesis testing is the question of distinguishing whether two unknown distributions are very similar, or significantly different. Classical tests, like the Chi-squared test or the Kolmogorov-Smirnov statistic, are optimal in the asymptotic regime, for fixed distributions as the sample sizes tend towards infinity. Nevertheless, in many modern settings—such as the analysis of customer, web logs, natural language processing, and genomics, despite the quantity of available data—the support sizes and complexity of the underlying distributions are far larger than the datasets, as evidenced by the fact that many phenomena are observed only a single time in the datasets, and the empirical distributions of the samples are poor representations of the true underlying distributions.[1] In such settings, we must understand these statistical tasks not only in the asymptotic regime (in which the amount of available data goes to infinity), but in the "undersampled" regime in which the dataset is significantly smaller than the size or complexity of the distribution in question. Surprisingly, despite an intense history of study by the statistics, information theory, and computer science communities, aspects of basic hypothesis testing and estimation questions–especially in the undersampled regime—remain unresolved, and require both new algorithms, and new analysis techniques.

---

[*]Supported in part by NSF CAREER Award CCF-1351108

[1]To give some specific examples, two recent independent studies [19, 26] each considered the genetic sequences of over 14,000 individuals, and found that rare variants are extremely abundant, with over 80% of mutations observed just once in the sample. A separate recent paper [16] found that the discrepancy in rare mutation abundance cited in different demographic modeling studies can largely be explained by discrepancies in the sample sizes of the respective studies, as opposed to differences in the actual distributions of rare mutations across demographics, highlighting the importance of improved statistical tests in this "undersampled" regime.

In this work, we examine the basic hypothesis testing question of deciding whether two unknown distributions over discrete supports are identical (or extremely similar), versus have total variation distance at least $\varepsilon$, for some specified parameter $\varepsilon > 0$. We consider (and largely resolve) this question in the extremely practically relevant setting of *unequal sample sizes*. Informally, taking $\varepsilon$ to be a small constant, we show that provided $p$ and $q$ are supported on at most $n$ elements, for any $\gamma \in [0, 1/3]$, the hypothesis test can be successfully performed (with high probability over the random samples) given samples of size $m_1 = \Theta(n^{2/3+\gamma})$ from $p$, and $m_2 = \Theta(n^{2/3-\gamma/2})$ from $q$, where $n$ is the size of the supports of the distributions $p$ and $q$. Furthermore, for every $\gamma$ in this range, this tradeoff between $m_1$ and $m_2$ is necessary, up to constant factors. Thus, our results smoothly interpolate between the known bounds of $\Theta(n^{2/3})$ on the sample size necessary in the setting where one is given two equal-sized samples [6, 9], and the bound of $\Theta(\sqrt{n})$ on the sample size in the setting in which the sample is drawn from one distribution and the other distribution is *known* to the algorithm [22, 29]. Throughout most of the regime of parameters, when $m_1 \ll m_2^2$, our algorithm is a natural extension of the algorithm proposed in [9], and is similar to the algorithm proposed in [3] except with the addition of a normalization term that seems crucial to obtaining our information theoretic optimality. In the extreme regime when $m_1 \approx n$ and $m_2 \approx \sqrt{n}$, our algorithm introduces an additional statistic which (we believe) is new. Our algorithm is relatively simple, and practically viable. In Section 4 we illustrate the efficacy of our approach on both synthetic data, and on the real-world problem of deducing whether two words are synonyms, based on a small sample of the bi-grams in which they occur.

We also note that, as pointed out in several related work [3, 12, 6], this hypothesis testing question has applications to other problems, such as estimating or testing the mixing time of Markov chains, and our results yield improved algorithms in these settings.

## 1.1 Related Work

The general question of how to estimate or test properties of distributions using fewer samples than would be necessary to actually learn the distribution, has been studied extensively since the late '90s. Most of the work has focussed on "symmetric" properties (properties whose value is invariant to relabeling domain elements) such as entropy, support size, and distance metrics between distributions (such as $\ell_1$ distance). This has included both algorithmic work (e.g. [4, 5, 7, 8, 10, 13, 20, 21, 27, 28, 29]), and results on developing techniques and tools for establishing lower bounds (e.g. [23, 30, 27]). See the recent survey by Rubinfeld for a more thorough summary of the developments in this area [24]).

The specific problem of "closeness testing" or "identity testing", that is, deciding whether two distributions, $p$ and $q$, are similar, versus have significant distance, has two main variants: the *one-unknown-distribution* setting in which $q$ is known and a sample is drawn from $p$, and the *two-unknown-distributions* settings in which both $p$ and $q$ are unknown and samples are drawn from both. We briefly summarize the previous results for these two settings.

In the one-unknown-distribution setting (which can be thought of as the limiting setting in the case that we have an arbitrarily large sample drawn from distribution $q$, and a relatively modest sized sample from $p$), initial work of Goldreich and Ron [12] considered the problem of testing whether $p$ is the uniform distribution over $[n]$, versus has distance at least $\varepsilon$. The tight bounds of $\Theta(\sqrt{n}/\varepsilon^2)$ were later shown by Paninski [22], essentially leveraging the birthday paradox and the intuition that, among distributions supported on $n$ elements, the uniform distribution maximizes the number of domain elements that will be observed once. Batu et al. [8] showed that, up to polylogarithmic factors of $n$, and polynomial factors of $\varepsilon$, this dependence was optimal for worst-case distributions over $[n]$. Recently, an "instance–optimal" algorithm and matching lower bound was shown: for any distribution $q$, up to constant factors, $\max\{\frac{1}{\varepsilon}, \varepsilon^{-2}\|q_{-\Theta(\varepsilon)}^{-\max}\|_{2/3}\}$ samples from $p$ are both necessary and sufficient to test $p = q$ versus $\|p - q\| \geq \varepsilon$, where $\|q_{-\Theta(\varepsilon)}^{-\max}\|_{2/3} \leq \|q\|_{2/3}$ is the 2/3-rd norm of the vector of probabilities of distribution $q$ after the maximum element has been removed, and the smallest elements up to $\Theta(\varepsilon)$ total mass have been removed. (This immediately implies the tight bounds that if $q$ is any distribution supported on $[n]$, $O(\sqrt{n}/\varepsilon^2)$ samples are sufficient to test its identity.)

The two-unknown-distribution setting was introduced to this community by Batu et al. [6]. The optimal sample complexity of this problem was recently determined by Chan et al. [9]: they showed

that $m = \Theta(n^{2/3}/\varepsilon^{4/3})$ samples are necessary and sufficient. In a slightly different vein, Acharya et al. [1, 2] recently considered the question of closeness testing with two unknown distributions from the standpoint of competitive analysis. They proposed an algorithm that performs the desired task using $O(s^{3/2} \operatorname{polylog} s)$ samples, and established a lower bound of $\Omega(s^{7/6})$, where $s$ represents the number of samples required to determine whether a set of samples were drawn from $p$ versus $q$, in the setting where $p$ and $q$ are explicitly known.

A natural generalization of this hypothesis testing problem, which interpolates between the two-unknown-distribution setting and the one-unknown-distribution setting, is to consider unequal sized samples from the two distributions. More formally, given $m_1$ samples from the distribution $p$, the *asymmetric closeness testing* problem is to determine how many samples, $m_2$, are required from the distribution $q$ such that the hypothesis $p = q$ versus $||p - q||_1 > \varepsilon$ can be distinguished with large constant probability (say 2/3). Note that the results of Chan et al. [9] imply that it is sufficient to consider $m_1 \geq \Theta(n^{2/3}/\varepsilon^{4/3})$. This problem was studied recently by Acharya et al. [3]: they gave an algorithm that given $m_1$ samples from the distribution $p$ uses $m_2 = O(\max\{\frac{n \log n}{\varepsilon^3 \sqrt{m_1}}, \frac{\sqrt{n \log n}}{\varepsilon^2}\})$ samples from $q$, to distinguish the two distributions with high probability. They also proved a lower bound of $m_2 = \Omega(\max\{\frac{\sqrt{n}}{\varepsilon^2}, \frac{n^2}{\varepsilon^4 m_1^2}\})$. There is a polynomial gap in these upper and lower bounds in the dependence on $n$, $\sqrt{m_1}$ and $\varepsilon$.

As a corollary to our main hypothesis testing result, we obtain an improved algorithm for testing the mixing time of a Markov chain. The idea of testing mixing properties of a Markov chain goes back to the work of Goldreich and Ron [12], which conjectured an algorithm for testing expansion of bounded-degree graphs. Their test is based on picking a random node and testing whether random walks from this node reach a distribution that is close to the uniform distribution on the nodes of the graph. They conjectured that their algorithm had $O(\sqrt{n})$ query complexity. Later, Czumaj and Sohler [11], Kale and Seshadhri [15], and Nachmias and Shapira [18] have independently concluded that the algorithm of Goldreich and Ron is provably a test for expansion property of graphs. Rapid mixing of a chain can also be tested using eigenvalue computations. Mixing is related to the separation between the two largest eigenvalues [25, 17], and eigenvalues of a dense $n \times n$ matrix can be approximated in $O(n^3)$ time and $O(n^2)$ space. However, for a sparse $n \times n$ symmetric matrix with $m$ nonzero entries, the same task can be achieved in $O(n(m + \log n))$ operations and $O(n + m)$ space. Batu et al. [6] used their $\ell_1$ distance test on the $t$-step distributions, to test mixing properties of Markov chains. Given a finite Markov chain with state space $[n]$ and transition matrix $\boldsymbol{P} = ((P(x, y)))$, they essentially show that one can estimate the mixing time $\tau_{mix}$ up to a factor of $\log n$ using $\tilde{O}(n^{5/3} \tau_{mix})$ queries to a *next node* oracle, which takes a state $x \in [n]$ and outputs a state $y \in [n]$ drawn from the distribution $P(x, \cdot)$. Such an oracle can often be simulated significantly more easily than actually computing the transition matrix $P(x, y)$.

We conclude this related work section with a comment on "robust" hypothesis testing and distance estimation. A natural hope would be to simply estimate $||p - q||$ to within some additive $\varepsilon$, which is a strictly more difficult task than distinguishing $p = q$ from $||p - q|| \geq \varepsilon$. The results of Valiant and Valiant [27, 28, 29] show that this problem is significantly more difficult than hypothesis testing: the distance can be estimated to additive error $\varepsilon$ for distributions supported on $\leq n$ elements using samples of size $O(n/\log n)$ (in both the setting where either one, or both distributions are unknown). Moreover, $\Omega(n/\log n)$ samples are information theoretically necessary, even if $q$ is the uniform distribution over $[n]$, and one wants to distinguish the case that $||p - q||_1 \leq \frac{1}{10}$ from the case that $||p - q||_1 \geq \frac{9}{10}$. Recall that the non-robust test of distinguishing $p = q$ versus $||p - q|| > 9/10$ requires a sample of size only $O(\sqrt{n})$. The exact worst-case sample complexity of distinguishing whether $||p - q||_1 \leq \frac{1}{n^c}$ versus $||p - q||_1 \geq \varepsilon$ is not well understood, though in the case of constant $\varepsilon$, up to logarithmic factors, the required sample size seems to scale linearly in the exponent between $n^{2/3}$ and $n$ as $c$ goes from 1/3 to 0.

## 1.2   Our results

Our main result resolves the minimax sample complexity of the closeness testing problem in the unequal sample setting, to constant factors, in terms of $n$, the support sizes of the distributions in question:

**Theorem 1.** *Given $m_1 \geq n^{2/3}/\varepsilon^{4/3}$ and $\varepsilon > n^{-1/12}$, and sample access to distributions $p$ and $q$ over $[n]$, there is an $O(m_1)$ time algorithm which takes $m_1$ independent draws from $p$ and $m_2 = O(\max\{\frac{n}{\sqrt{m_1}\varepsilon^2}, \frac{\sqrt{n}}{\varepsilon^2}\})$ independent draws from $q$, and with probability at least 2/3 distinguishes whether*

$$||p - q||_1 \leq O\left(\frac{1}{m_2}\right) \quad versus \quad ||p - q||_1 \geq \varepsilon. \tag{1}$$

*Moreover, given $m_1$ samples from $p$, $\Omega(\max\{\frac{n}{\sqrt{m_1}\varepsilon^2}, \frac{\sqrt{n}}{\varepsilon^2}\})$ samples from $q$ are information-theoretically necessary to distinguish $p = q$ from $||p - q||_1 \geq \varepsilon$ with any constant probability bounded below by $1/2$.*

The lower bound in the above theorem is proved using the machinery developed in Valiant [30], and "interpolates" between the $\Theta(\sqrt{n}/\varepsilon^2)$ lower bound in the one-unknown-distribution setting of testing uniformity [22] and the $\Theta(n^{2/3}/\varepsilon^{4/3})$ lower bound in the setting of equal sample sizes from two unknown distributions [9]. The algorithm establishing the upper bound involves a re-weighted version of a statistic proposed in [9], and is similar to the algorithm proposed in [3] modulo the addition of a normalizing term, which seems crucial to obtaining our tight results. In the extreme regime when $m_1 \approx n$ and $m_2 \approx \sqrt{n}/\varepsilon^2$, we incorporate an additional statistic that has not appeared before in the literature.

As an application of Theorem 1 in the extreme regime when $m_1 \approx n$, we obtain an improved algorithm for estimating the mixing time of a Markov chain:

**Corollary 1.** *Consider a finite Markov chain with state space $[n]$ and a next node oracle; there is an algorithm that estimates the mixing time, $\tau_{mix}$, up to a multiplicative factor of $\log n$, that uses $\tilde{O}(n^{3/2}\tau_{mix})$ time and queries to the next node oracle.*

Concurrently to our work, Hsu et al. [14] considered the question of estimating the mixing time based on a single sample path (as opposed to our model of a sampling oracle). In contrast to our approach via hypothesis testing, they considered the natural spectral approach, and showed that the mixing time can be approximated, up to logarithmic factors, given a path of length $\tilde{O}(\tau_{mix}^3/\pi_{\min})$, where $\pi_{\min}$ is the minimum probability of a state under the stationary distribution. Hence, if the stationary distribution is uniform over $n$ states, this becomes $\tilde{O}(n\tau_{mix}^3)$. It remains an intriguing open question whether one can simultaneously achieve both the linear dependence on $\tau_{mix}$ of our results and the linear dependence on $1/\pi_{\min}$ or the size of the state space, $n$, as in their results.

### 1.3 Outline

We begin by stating our testing algorithm, and describe the intuition behind the algorithm. The formal proof of the performance guarantees of the algorithm require rather involved bounds on the moments of various parameters, and are provided in the supplementary material. We also defer the entirety of the matching information theoretic lower bounds to the supplementary material, as the techniques may not appeal to as wide an audience as the algorithmic portion of our work. The application of our testing results to the problem of testing or estimating the mixing time of a Markov chain is discussed in Section 3. Finally, Section 4 contains some empirical results, suggesting that the statistic at the core of our testing algorithm performs very well in practice. This section contains both results on synthetic data, as well as an illustration of how to apply these ideas to the problem of estimating the semantic similarity of two words based on samples of the $n$-grams that contain the words in a corpus of text.

## 2 Algorithms for $\ell_1$ Testing

In this section we describe our algorithm for $\ell_1$ testing with unequal samples. This gives the upper bound in Theorem 1 on the sample sizes necessary to distinguish $p = q$ from $||p - q||_1 \geq \varepsilon$. For clarity and ease of exposition, in this section we consider $\varepsilon$ to be some absolute constant, and supress the dependency on $\varepsilon$. The slightly more involved algorithm that also obtains the optimal dependency on the parameter $\varepsilon$ is given in the supplementary material.

We begin by presenting the algorithm, and then discuss the intuition for the various steps.

---

**Algorithm 1** The Closeness Testing Algorithm

---

Suppose $\varepsilon = \Omega(1)$ and $m_1 = O(n^{1-\gamma})$ for some $\gamma \geq 0$. Let $S_1, S_2$ denote two independent sets of $m_1$ samples drawn from $p$ and let $T_1, T_2$ denote two independent sets of $m_2$ samples drawn from $q$. We wish to test $p = q$ versus $||p - q||_1 > \varepsilon$.

- Let $b = C_0 \frac{\log n}{m_2}$, for an absolute constant $C_0$, and define the set

  $B = \{i \in [n] : \frac{X_i^{S_1}}{m_1} > b\} \cup \{i \in [n] : \frac{Y_i^{T_1}}{m_2} > b\}$, where $X_i^{S_1}$ denotes the number of occurrences of $i$ in $S_1$, and $Y_i^{T_1}$ denotes the number of occurrences of $i$ in $T_1$.

- Let $X_i$ denote the number of occurrences of element $i$ in $S_2$, and $Y_i$ denote the number of occurrences of element $i$ in $T_2$:

1. Check if

$$\sum_{i \in B} \left| \frac{X_i}{m_1} - \frac{Y_i}{m_2} \right| \leq \varepsilon/6. \tag{2}$$

2. Check if

$$Z := \sum_{i \in [n] \setminus B} \frac{(m_2 X_i - m_1 Y_i)^2 - (m_2^2 X_i + m_1^2 Y_i)}{X_i + Y_i} \leq C_\gamma m_1^{3/2} m_2, \tag{3}$$

   for an appropriately chosen constant $C_\gamma$ (depending on $\gamma$).

3. If $\gamma \geq 1/9$:
   - If (2) and (3) hold, then ACCEPT. Otherwise, REJECT.

4. Otherwise, if $\gamma < 1/9$ :
   - Check if

$$R := \sum_{i \in [n] \setminus B} \frac{\mathbf{1}\{Y_i = 2\}}{X_i + 1} \leq C_1 \frac{m_2^2}{m_1}, \tag{4}$$

   where $C_1$ is an appropriately chosen absolute constant.
   - REJECT if there exists $i \in [n]$ such that $Y_i \geq 3$ and $X_i \leq C_2 \frac{m_1}{m_2 n^{1/3}}$, where $C_2$ is an appropriately chosen absolute constant.
   - If (2), (3), and (4) hold, then ACCEPT. Otherwise, REJECT.

---

The intuition behind the above algorithm is as follows: with high probability, all elements in the set $B$ satisfy either $p_i > b/2$, or $q_i > b/2$ (or both). Given that these elements are "heavy", their contribution to the $\ell_1$ distance will be accurately captured by the $\ell_1$ distance of their empirical frequencies (where these empirical frequencies are based on the second set of samples, $S_2, T_2$).

For the elements that are not in set $B$—the "light" elements—their empirical frequencies will, in general, not accurately reflect their true probabilities, and hence the distance between the empirical distributions of the "light" elements will be misleading. The $Z$ statistic of Equation 3 is designed specifically for this regime. If the denominator of this statistic were omitted, then this would give an estimator for the squared $\ell_2$ distance between the distributions (scaled by a factor of $m_1^2 m_2^2$). To see this, note that if $p_i$ and $q_i$ are small, then $Binomial(m_1, p_i) \approx Poisson(m_1 p_i)$ and $Binomial(m_2, q_i) \approx Poisson(m_2 q_i)$; furthermore, a simple calculation yields that if $X_i \leftarrow Poisson(m_1 p_i)$ and $Y_i \leftarrow Poisson(m_2 q_i)$, then $\mathbb{E}\left[(m_2 X_i - m_1 Y_i)^2 - (m_2^2 X_i + m_1^2 Y_i)\right] = m_1^2 m_2^2 (p - q)^2$. The normalization by $X_i + Y_i$ "linearizes" the $Z$ statistic, essentially turning the squared $\ell_2$ distance into an estimate of the $\ell_1$ distance between light elements of the two distributions. Similar results can possibly be obtained using other linear functions of $X_i$ and $Y_i$ in the denominator, though we note that the "obvious" normalizing factor of $X_i + \frac{m_1}{m_2} Y_i$ does not seem to work theoretically, and seems to have extremely poor performance in practice.

For the extreme case (corresponding to $\gamma < 1/9$) where $m_1 \approx n$ and $m_2 \approx \sqrt{n}/\varepsilon^2$, the statistic $Z$ might have a prohibitively large variance; this is essentially due to the "birthday paradox" which might cause a constant number of rare elements (having probability $O(1/n)$ to occur twice in a sample of size $m_2 \approx \sqrt{n}/\varepsilon^2$). Each such element will contribute $\Omega(m_1^2) \approx n^2$ to the $Z$ statistic,

and hence the variance can be $\approx n^4$. The statistic $R$ of Equation (4) is tailored to deal with these cases, and captures the intuition that we are more tolerant of indices $i$ for which $Y_i = 2$ if the corresponding $X_i$ is larger. It is worth noting that one can also define a natural analog of the $R$ statistic corresponding to the indices $i$ for which $Y_i = 3$, etc., using which the robustness parameter of the test can be improved. The final check—ensuring that in this regime with $m_1 \gg m_2$ there are no elements for which $Y_i \geq 3$ but $X_i$ is small—rules out the remaining sets of distributions, $p, q$, for which the variance of the $Z$ statistic is intolerably large.

Finally, we should emphasize that the crude step of using two independent batches of samples— the first to obtain the partition of the domain into "heavy" and "light" elements, and the second to actually compute the statistics, is for ease of analysis. As our empirical results of Section 4 suggest, for practical applications one may want to use only the $Z$-statistic of (3), and one certainly should not "waste" half the samples to perform the "heavy"/"light" partition.

## 3 Estimating Mixing Times in Markov Chains

The basic hypothesis testing question of distinguishing identical distributions from those with significant $\ell_1$ distance can be employed for several other practically relevant tasks. One example is the problem of estimating the mixing time of Markov chains.

Consider a finite Markov chain with state space $[n]$, transition matrix $\boldsymbol{P} = ((P(x, y)))$, with stationary distribution $\pi$. The *t-step distribution starting at the point* $x \in [n]$, $P_x^t(\cdot)$ is the probability distribution on $[n]$ obtained by running the chain for $t$ steps starting from $x$.

**Definition 1.** *The $\varepsilon$-mixing time of a Markov chain with transition matrix $\boldsymbol{P} = ((P(x, y)))$ is defined as* $t_{\mathrm{mix}}(\varepsilon) := \inf \left\{ t \in [n] : \sup_{x \in [n]} \frac{1}{2} \sum_{y \in [n]} |P_x^t(y) - \pi(y)| \leq \varepsilon \right\}$.

**Definition 2.** *The average $t$-step distribution of a Markov chain $\boldsymbol{P}$ with $n$ states is the distribution $\overline{P}^t = \frac{1}{n} \sum_{x \in [n]} P_x^t$, that is, the distribution obtained by choosing $x$ uniformly from $[n]$ and walking $t$ steps from the state $x$.*

The connection between closeness testing and testing whether a Markov chain is close to mixing was first observed by Batu et al. [6], who proposed testing the $\ell_1$ difference between distributions $P_x^{t_0}$ and $\overline{P}^{t_0}$, for every $x \in [n]$. The algorithm leveraged their *equal* sample-size hypothesis testing results, drawing $\tilde{O}(n^{2/3} \log n)$ samples from both the distributions $P_x^{t_0}$ and $\overline{P}^{t_0}$. This yields an overall running time of $\tilde{O}(n^{5/3} t_0)$.

Here, we note that our *unequal* sample-size hypothesis testing algorithm can yield an improved runtime. Since the distribution $\overline{P}^{t_0}$ is independent of the starting state $x$, it suffices to take $\tilde{O}(n)$ samples from $\overline{P}^{t_0}$ once and $\tilde{O}(\sqrt{n})$ samples from $P_x^t$, for every $x \in [n]$. This results in a query and runtime complexity of $\tilde{O}(n^{3/2} t_0)$. We sketch this algorithm below.

---

**Algorithm 2** Testing for Mixing Times in Markov Chains

---

Given $t_0 \in \mathbb{R}$ and a finite Markov chain with state space $[n]$ and transition matrix $\boldsymbol{P} = ((P(x, y)))$, we wish to test

$$H_0 : t_{\mathrm{mix}} \left( O \left( \frac{1}{\sqrt{n}} \right) \right) \leq t_0, \quad \text{versus} \quad H_1 : t_{\mathrm{mix}}(1/4) > t_0. \tag{5}$$

1. Draw $O(\log n)$ samples $S_1, \ldots, S_{O(\log n)}$, each of size $\mathrm{Pois}(C_1 n)$ from the average $t_0$-step distribution.

2. For each state $x \in [n]$ we will distinguish whether $||P_x^{t_0} - \overline{P}^{t_0}||_1 \leq O(\frac{1}{\sqrt{n}})$, versus $||P_x^{t_0} - \overline{P}^{t_0}||_1 > 1/4$, with probability of error $\ll 1/n$. We do this by running $O(\log n)$ runs of Algorithm 1, with the $i$-th run using $S_i$ and a fresh set of $\mathrm{Pois}(O(\sqrt{n}))$ samples from $P_x^t$.

3. If all $n$ of the $\ell_1$ closeness testing problems are accepted, then we ACCEPT $H_0$.

---

The above testing algorithm can be leveraged to *estimate* the mixing time of a Markov chain, via the basic observation that if $t_{\mathrm{mix}}(1/4) \leq t_0$, then for any $\varepsilon$, $t_{\mathrm{mix}}(\varepsilon) \leq \frac{\log \varepsilon}{\log 1/2} t_0$, and thus $t_{\mathrm{mix}}(1/\sqrt{n}) \leq 2 \log n \cdot t_{\mathrm{mix}}(1/4)$. Because $t_{\mathrm{mix}}(1/4)$ and $t_{\mathrm{mix}}(O(1/\sqrt{n}))$ differ by at most a factor of $\log n$, by applying Algorithm 2 for a geometrically increasing sequence of $t_0$'s, and repeating each test $O(\log t_0 + \log n)$ times, one obtains Corollary 1, restated below:

**Corollary 1** *For a finite Markov chain with state space* $[n]$ *and a* next node oracle*, there is an algorithm that estimates the mixing time,* $\tau_{mix}$*, up to a multiplicative factor of* $\log n$*, that uses* $\tilde{O}(n^{3/2}\tau_{mix})$ *time and queries to the next node oracle.*

## 4 Empirical Results

Both our formal algorithms and the corresponding theorems involve some unwieldy constant factors (that can likely be reduced significantly). Nevertheless, in this section we provide some evidence that the statistic at the core of our algorithm can be fruitfully used in practice, even for surprisingly small sample sizes.

### 4.1 Testing similarity of words

An extremely important primitive in natural language processing is the ability to estimate the *semantic similarity* of two words. Here, we show that the $Z$ statistic, $Z = \sum_i \frac{(m_2 X_i - m_1 Y_i)^2 - (m_2^2 X_i + m_1^2 Y_i)}{m_1^{3/2} m_2 (X_i + Y_i)}$, which is the core of our testing algorithm, can accurately distinguish whether two words are very similar based on surprisingly small samples of the contexts in which they occur. Specifically, for each pair of words, $a, b$ that we consider, we select $m_1$ random occurrences of $a$ and $m_2$ random occurrences of word $b$ from the Google books corpus, using the Google Books Ngram Dataset.[2] We then compare the sample of words that follow $a$ with the sample of words that follow $b$. Henceforth, we refer to these as samples of the set of bi-grams involving each word.

Figure 1(a) illustrates the $Z$ statistic for various pairs of words that range from rather similar words like "smart" and "intelligent", to essentially identical word pairs such as "grey" and "gray" (whose usage differs mainly as a result of historical variation in the preference for one spelling over the other); the sample size of bi-grams containing the first word is fixed at $m_1 = 1,000$, and the sample size corresponding to the second word varies from $m_2 = 50$ through $m_2 = 1,000$. To provide a frame of reference, we also compute the value of the statistic for independent samples corresponding to the same word (i.e. two different samples of words that follow "wolf"); these are depicted in red. For comparison, we also plot the total variation distance between the empirical distributions of the pair of samples, which does not clearly differentiate between pairs of identical words, versus different words, particularly for the smaller sample sizes.

One subtle point is that the issue with using the empirical distance between the distributions goes beyond simply not having a consistent reference point. For example, let $X$ denote a large sample of size $m_1$ from distribution $p$, $X'$ denote a small sample of size $m_2$ from $p$, and $Y$ denote a small sample of size $m_2$ from a different distribution $q$. It is tempting to hope that the empirical distance between $X$ and $X'$ will be smaller than the empirical distance between $X$ and $Y$. As Figure 1(b) illustrates, this is not always the case, even for natural distributions: for the specific example illustrated in the figure, over much of the range of $m_2$, the empirical distance between $X$ and $X'$ is indistinguishable from that of $X$ and $Y$, though the $Z$ statistic easily discerns that these distributions are very different.

This point is further emphasized in Figure 2, which depicts this phenomena in the synthetic setting where $p = \mathrm{Unif}[n]$ is the uniform distribution over $n$ elements, and $q$ is the distribution whose elements have probabilities $(1 \pm \varepsilon)/n$, for $\varepsilon = 1/2$. The second and fourth plots represent the probability that the distance between two empirical distributions of samples from $p$ is smaller than the distance between the empirical distributions of the samples from $p$ and $q$; the first and third plots represent the analogous probability involving the $Z$ statistic. The first two plots correspond to $n = 1,000$ and the last two correspond to $n = 50,000$. In all plots, we consider a pair of samples of respective sizes $m_1$ and $m_2$, as $m_1$ and $m_2$ range between $\sqrt{n}$ and $n$.

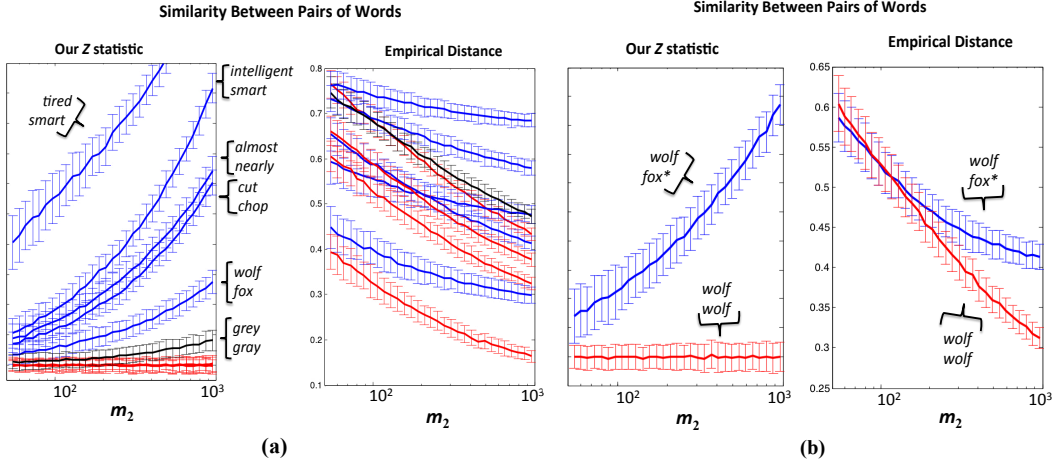

**(a)**                                            **(b)**

Figure 1: (a) Two measures of the similarity between words, based on samples of the bi-grams containing each word. Each line represents a pair of words, and is obtained by taking a sample of $m_1 = 1,000$ bi-grams containing the first word, and $m_2 = 50, \ldots, 1,000$ bi-grams containing the second word, where $m_2$ is depicted along the $x$-axis in logarithmic scale. In both plots, the red lines represent pairs of identical words (e.g. "wolf/wolf","almost/almost",...). The blue lines represent pairs of similar words (e.g. "wolf/fox", "almost/nearly",...), and the black line represents the pair "grey/gray" whose distribution of bi-grams differ because of historical variations in preference for each spelling. Solid lines indicate the average over 200 trials for each word pair and choice of $m_2$, with error bars of one standard deviation depicted. The left plot depicts our statistic, which clearly distinguishes identical words, and demonstrates some intuitive sense of semantic distance. The right plot depicts the total variation distance between the empirical distributions—which does not successfully distinguish the identical words, given the range of sample sizes considered. The plot would not be significantly different if other distance metrics between the empirical distributions, such as f-divergence, were used in place of total variation distance. Finally, note the extremely uniform magnitudes of the error bars in the left plot, as $m_2$ increases, which is an added benefit of the $X_i + Y_i$ normalization term in the $Z$ statistic. (b) Illustration of how the empirical distance can be misleading: here, the empirical distance between the distributions of samples of bi-grams for "wolf/wolf" is indistinguishable from that for the pair "wolf/fox*" over much of the range of $m_2$; nevertheless, our statistic clearly discerns that these are significantly different distributions. Here, "fox*" denotes the distribution of bi-grams whose first word is "fox", restricted to only the most common 100 bi-grams.

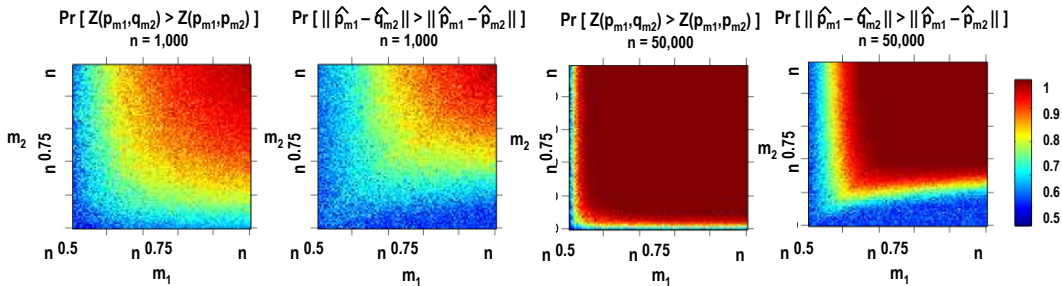

Figure 2: The first and third plot depicts the probability that the $Z$ statistic applied to samples of sizes $m_1, m_2$ drawn from $p = Unif[n]$ is smaller than the $Z$ statistic applied to a sample of size $m_1$ drawn from $p$ and $m_2$ drawn from $q$, where $q$ is a perturbed version of $p$ in which all elements have probability $(1 \pm 1/2)/n$. The second and fourth plots depict the probability that empirical distance between a pair of samples (of respective sizes $m_1, m_2$) drawn from $p$ is less than the empirical distribution between a sample of size $m_1$ drawn from $p$ and $m_2$ drawn from $q$. The first two plots correspond to $n = 1,000$ and the last two correspond to $n = 50,000$. In all plots, $m_1$ and $m_2$ range between $\sqrt{n}$ and $n$ on a logarithmic scale. In all plots the colors depict the average probability based on 100 trials.

## Footnotes

[2]The Google Books Ngram Dataset is freely available here: `http://storage.googleapis.com/books/ngrams/books/datasetsv2.html`

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
