[Supplementary Material]

# Testing Closeness With Unequal Sized Samples: Supplementary Material

This supplementary material is organized as follows. Section A describes the extension of Algorithm 3 that obtains the optimal dependence on the error parameter $\varepsilon$ (in addition to the optimal dependence on $n, m_1$, and $m_2$. Our description is split into two parts, corresponding to the regime in which $m_1 \leq n^{1-\gamma}$ for some positive constant $\gamma$, and the extreme regime where $m_1 \approx n$ and $m_2 \approx \sqrt{n}/\varepsilon^2$. We conclude Section A with a high level overview of the proof approach. Sections B, C, and D contain the formal proof of the analysis of the algorithms, proving the upper bounds of Theorem 1. Finally, in Section E we prove our information theoretic lower bounds, establishing the optimality of our testing algorithms.

Throughout this supplementary material, in both the description and analysis of the algorithms, and in the lower bound section, we work in the "Poissonized" setting, where we assume that we have access to $\mathrm{Pois}(m_1)$ samples from distribution $p$, and $\mathrm{Pois}(m_2)$ samples drawn distribution $q$. This assumption that the sample size is a random variable renders the number of occurrences of different domain elements independent. Because $\mathrm{Pois}(\lambda)$ is tightly concentrated about its expectation, both the upper and lower bounds on the sample complexities proved in this "Poissonized" setting also hold (up to factors of $1 \pm o(1)$) in the setting in which one obtains samples of a fixed size.

# A  Algorithms for $\ell_1$ Testing

To get the optimal dependence on $\varepsilon$, Algorithm 1 needs to be slightly modified. Algorithm 3 below gives the optimal sample complexity in the non-extreme case, for any $\varepsilon \geq n^{-\frac{1}{12}}$. In this case, a separate check needs to done to handle elements in the "medium set" $M$ defined below. Note that when $\varepsilon$ is a constant, the frequencies of the elements assigned to the medium set only differ by a constant factor and we can absorb them in either $B$ or $H$, hence recovering Algorithm 1.

---

**Algorithm 3** Asymmetric Closeness Testing: Non-Extreme Case

---

Suppose $m_1 = O\big((n/\varepsilon^2)^{1-\gamma}\big) \leq n$ for some $\gamma > 0$. Let $S_1, S_2$ denote two independent sets of $\mathrm{Pois}(m_1)$ samples from $p$ and let $T_1, T_2$ denote two independent sets of $\mathrm{Pois}(m_2)$ samples drawn from $q$. We wish to test $p = q$ versus $||p - q||_1 > \varepsilon$.

- Let $b = \frac{256 \log n}{\varepsilon^2 m_2}$, and $b' = \frac{256 \log n}{m_2}$, and let $X_i^{S_1}$ denote the number of occurrences of $i$ in $S_1$, and $Y_i^{T_1}$ denote the number of occurrences of $i$ in $T_1$.

- Define the "heavy" set $B = \{i \in [n] : \frac{X_i^{S_1}}{m_1} > b\} \cup \{i \in [n] : \frac{Y_i^{T_1}}{m_2} > b\}$.

- Define the "medium" set $M = \left\{ i \in [n] : b' \leq \max\{\frac{X_i^{S_1}}{m_1}, \frac{Y_i^{T_1}}{m_2}\} \leq b \right\}$.

- Define the "light" set $H = [m] \setminus (B \cup M)$.

- Let $X_i$ denote the number of occurrences of element $i$ in $S_2$, and $Y_i$ denote the number of occurrences of element $i$ in $T_2$:

1. Check if
$$V_B := \sum_{i \in B} V_i := \sum_{i \in B} \left| \frac{X_i}{m_1} - \frac{Y_i}{m_2} \right| \leq \varepsilon/6. \qquad (6)$$

2. Check if
$$W_M := \sum_{i \in M} W_i := \sum_{i \in M} (m_2 X_i - m_1 Y_i)^2 - (m_2^2 X_i + m_1^2 Y_i) \leq \frac{\varepsilon^2 m_1^2 m_2 \log n}{2}. \qquad (7)$$

3. Check if
$$Z_H := \sum_{i \in H} Z_i := \sum_{i \in H} \frac{(m_2 X_i - m_1 Y_i)^2 - (m_2^2 X_i + m_1^2 Y_i)}{X_i + Y_i} \leq C_\gamma m_1^{3/2} m_2, . \qquad (8)$$

    Where $C_\gamma$ is an appropriately chosen absolute constant, dependent on $\gamma$.

4. If (6), (7), and (8) hold, then ACCEPT. Otherwise, REJECT.

---

The following proposition characterizes the performance of this algorithm, establishing the upper bounds of Theorem 1 in the non-extreme range of parameters with $m_1 < n^{1-\gamma}$. The proof is given in Sections B and C.

**Proposition 1.** *Suppose* $m_1 = O\big((n/\varepsilon^2)^{1-\gamma}\big) \leq n$ *for some* $\gamma > 0$, *and* $\varepsilon > n^{-1/12}$. *Then Algorithm 3 takes* $\Theta(m_1)$ *samples from* $p$ *and* $O(\max\{\frac{n}{\sqrt{m_1}\varepsilon^2}, \frac{\sqrt{n}}{\varepsilon^2}\})$ *samples from* $q$, *and with probability at least 2/3 distinguishes whether* $p = q$ *versus* $||p - q||_1 \geq \varepsilon$.

As mentioned earlier, for the extreme case, that is, $m_1 \approx n$ and $m_2 \approx \sqrt{n}$, the re-weighted statistic $Z_H$ may have an unacceptably large variance, necessitating a modification to the algorithm in this extreme case. The statistic $R$ introduced below (9) is tailored to deal with these cases.

---

**Algorithm 4** Asymmetric Closeness Testing: Extreme Case

---

Suppose $m_1 = \Omega\big((n/\varepsilon^2)^{8/9+\gamma}\big)$ for some $\gamma > 0$. Let $S_1, S_2$ denote two independent sets of $\mathrm{Pois}(m_1)$ samples from $p$ and let $T_1, T_2$ denote two independent sets of $\mathrm{Pois}(m_2)$ samples drawn from $q$. We wish to test $p = q$ versus $||p - q||_1 > \varepsilon$.

- Define $b, b', B, M, H$ as in Algorithm 3.
- Let $X_i$ denote the number of occurrences of element $i$ in $S_2$, and $Y_i$ denote the number of occurrences of element $i$ in $T_2$:

1. REJECT if there exists $i \in [n]$ such that $Y_i \geq 3$ and $X_i \leq \frac{m_1 \varepsilon^{2/3}}{10 m_2 n^{1/3}}$.

2. Check if

$$R_H := \sum_{i \in H} \frac{\mathbf{1}\{Y_i = 2\}}{X_i + 1} \leq C_1 \frac{m_2^2}{m_1}, \tag{9}$$

   where $C_1$ is an appropriately chosen absolute constant.

3. If step (1) is not rejected and (6), (7), (8), and (9) are satisfied, then ACCEPT. Otherwise, REJECT.

---

The following proposition characterizes the performance of this algorithm, establishing the upper bounds of Theorem 1 in the extreme range of parameters with $m_1 \approx n$. The proof is given in Sections B and D.

**Proposition 2.** *Suppose $m_1 = \Omega\big((n/\varepsilon^2)^{8/9+\gamma}\big)$ for some $\gamma > 0$ and $\varepsilon > n^{-1/12}$. Then Algorithm 4 takes $\Theta(m_1)$ samples from $p$ and $O(\max\{\frac{n}{\sqrt{m_1}\varepsilon^2}, \frac{\sqrt{n}}{\varepsilon^2}\})$ samples from $q$, and with probability at least 2/3 distinguishes whether $p = q$ versus $||p - q||_1 \geq \varepsilon$.*

### A.1 Proof Overview

At a high level, the analysis of the algorithms proceeds as follows: We first establish that, with high probability over the first set of samples, $S_1, T_1$, the sets $B, M, H$ successfully partition the elements in the "heavy", "medium", and "light" sets. This portion of the proof is completely standard and will follow from a union bound over Chernoff bounds and bounds on the tails of the Poisson distribution. The proofs of Propositions 1 and 2 will then proceed by arguing that, with high probability over the randomness of the second set of samples, $S_2, T_2$, the algorithms will be successful, provided that the sets $B, M, H$, were a reasonable partition. For this portion of the proofs, we need to ensure that the likely range of values that the various statistics ($V, W, Z$, and $R$) take in the equality setting $p = q$ are essentially disjoint from the range of values that the statistics would take in the setting where $||p - q|| \geq \varepsilon$. This argument applies Chebyshev's inequality, and its higher moment analogues. To enable this analysis, in Section B we first establish various bounds on the moments of these statistics, which we leverage throughout the remainder of the proof.

## B   Expectation and Variance Bounds

Before beginning the analysis of the above algorithms we need bounds on the expectation and variance of the different statistics used in the above algorithms. Throughout this section, fix any set $A \subseteq [n]$, and let $X_i$ denote the number of occurrences of the $i$-th domain element in set $S_2$—a set of $\mathrm{Pois}(m_1)$ samples from distribution $p$, and analogously let $Y_i$ denote the number of occurrences of the $i$-th domain element in set $T_2$—a set of $\mathrm{Pois}(m_2)$ samples from distribution $q$. Throughout this section, we bound the moments of the following statistics:

- $V_A = \sum_{i \in A} V_i = \sum_{i \in A} \left| \frac{X_i}{m_1} - \frac{Y_i}{m_2} \right|$.
- $W_A = \sum_{i \in A} W_i = \sum_{i \in A} \left( (m_2 X_i - m_1 Y_i)^2 - (m_2^2 X_i + m_1^2 Y_i) \right)$.
- $Z_A = \sum_{i \in A} Z_i = \sum_{i \in A} \frac{(m_2 X_i - m_1 Y_i)^2 - (m_2^2 X_i + m_1^2 Y_i)}{X_i + Y_i}$.

## B.1 Expectation and Variance of $V_A$

**Lemma 1.** *For any fixed set $A \subseteq [n]$*

$$\sum_{i \in A} |p_i - q_i| \leq \mathbb{E}[V_A] \leq \sum_{i \in A} |p_i - q_i| + \left(\frac{|A|}{m_1} + \frac{|A|}{m_2}\right)^{\frac{1}{2}} \leq \sum_{i \in A} |p_i - q_i| + \left(\frac{2|A|}{m_2}\right)^{\frac{1}{2}}, \quad (10)$$

*and*

$$\text{Var}[V_A] \leq \frac{1}{m_1} + \frac{1}{m_2}. \quad (11)$$

*Proof.* For the lower bound on the expectation, note that $\mathbb{E}\left[\left|\frac{X_i}{m_1} - \frac{Y_i}{m_2}\right|\right] \geq \left|\mathbb{E}\left[\frac{X_i}{m_1} - \frac{Y_i}{m_2}\right]\right| = |p_i - q_i|$.

To prove the upper bound, observe that

$$\mathbb{E}[V_i^2] = \frac{p_i}{m_1} + \frac{q_i}{m_2} + (p_i - q_i)^2.$$

By the Cauchy-Schwarz inequality,

$$\mathbb{E}\left[\sum_{i \in A} V_i\right] \leq \sum_{i \in A} \mathbb{E}[V_i^2]^{\frac{1}{2}} \leq \sum_{i \in A} |p_i - q_i| + \sum_{i \in A} \left(\frac{p_i}{m_1} + \frac{q_i}{m_2}\right)^{\frac{1}{2}}$$

$$\leq \sum_{i \in A} |p_i - q_i| + \left(\frac{|A|}{m_1} + \frac{|A|}{m_2}\right)^{\frac{1}{2}}. \quad (12)$$

Finally, $\text{Var}[V_A] = \sum_{i \in A}(\mathbb{E}[V_i^2] - \mathbb{E}[V_i]^2) \leq \frac{\sum_{i \in A} p_i}{m_1} + \frac{\sum_{i \in A} q_i}{m_2} \leq \frac{1}{m_1} + \frac{1}{m_2}$. $\qquad \square$

## B.2 Expectation and Variance of $W_A$

For $A \subseteq [n]$, define $W_A = \sum_{i \in A} W_i = \sum_{i \in A}(m_2 X_i - m_1 Y_i)^2 - (m_2^2 X_i + m_1^2 Y_i)$. Using the facts that $X_i \sim \text{Pois}(m_1 p_i)$ and $Y_i \sim \text{Pois}(m_2 q_i)$ and plugging in the expressions for the moments of Poissons, the next lemma follows immediately:

**Lemma 2.** *For any $A \subseteq [n]$, $W_A/(m_1^2 m_2^2)$ is an unbiased estimate of $\sum_{i \in A}(p_i - q_i)^2$. Namely,*

$$\mathbb{E}[W_A] = m_1^2 m_2^2 \sum_{i \in A}(p_i - q_i)^2, \quad (13)$$

*Moreover,*

$$\text{Var}[W_A] = 2m_1^2 m_2^2 \sum_{i \in A} z_i^2 + 4m_1^3 m_2^3 \sum_{i \in A} z_i(p_i - q_i)^2, \quad (14)$$

*where $z_i = m_2 p_i + m_1 q_i$.*

## B.3 Moments of $Z_A$

Recall that

$$Z_i := \frac{(m_2 X_i - m_1 Y_i)^2 - (m_2^2 X_i + m_1^2 Y_i)}{X_i + Y_i},$$

and for $A \subseteq [n]$, $Z_A := \sum_{i \in A} Z_i$. We show that if $p = q$, then $\mathbb{E}[\sum_{i \in A} Z_i] = 0$, and otherwise, we give a lower bound on the expectation of the sum:

**Lemma 3.** *If $p = q$, then $\mathbb{E}[\sum_{i \in A} Z_i] = 0$, and otherwise, $\mathbb{E}[\sum_{i \in A} Z_i] \geq \frac{m_1^2 m_2^2 (\sum_{i \in A} |p_i - q_i|)^2}{4n + m_1 + m_2}$.*

*Proof.* Conditioned on the denominator,

$$X_i \Big| X_i + Y_i = \sigma \sim \mathrm{Bin}\left(\sigma, \frac{m_1 p_i}{m_1 p_i + m_2 q_i}\right).$$

Set $\beta_i = \frac{m_1 p_i}{m_1 p_i + m_2 q_i}$. Then using binomial moments we get,

$$
\begin{aligned}
\mathbb{E}[(m_2 X_i - m_1 Y_i)^2 | X_i + Y_i = \sigma] &= \sigma \beta_i (1 - \beta_i)(m_1 + m_2)^2 + \sigma^2 (m_2 \beta_i - m_1(1 - \beta_i))^2 \\
&= (m_1 + m_2)^2 \left(\sigma \beta_i (1 - \beta_i) + \sigma^2 \left(\frac{m_1}{m_1 + m_2} - \beta_i\right)^2\right) \quad (15)
\end{aligned}
$$

Similarly,

$$
\begin{aligned}
\mathbb{E}[m_2^2 X_i + m_1^2 Y_i | X_i + Y_i = \sigma] &= m_1^2 \sigma + (m_2^2 - m_1^2)\mathbb{E}[X_i | X_i + Y_i = \sigma] \\
&= m_1^2 \sigma + (m_2^2 - m_1^2)\sigma \beta_i
\end{aligned}
$$

Therefore, the conditional expectation of the numerator is

$$
\begin{aligned}
\mathbb{E}\left[(m_2 X_i - m_1 Y_i)^2 - (m_2^2 X_i + m_1^2 Y_i) \Big| X_i + Y_i = \sigma\right] &= (m_1 + m_2)^2 \sigma(\sigma - 1)\left(\frac{m_1}{m_1 + m_2} - \beta_i\right)^2 \\
&= \sigma(\sigma - 1)\left(\frac{m_1 m_2 (q_i - p_i)}{m_1 p_i + m_2 q_i}\right)^2. \quad (16)
\end{aligned}
$$

This implies

$$\mathbb{E}\left[\sum_{i \in A} Z_i / m_1^2 m_2^2\right] = \sum_{i \in A} \frac{(q_i - p_i)^2}{z_i}\left(1 - \frac{1 - e^{-z_i}}{z_i}\right),$$

where $z_i = m_1 p_i + m_2 q_i$. This implies that the expectation of the sum is zero if $p = q$.

For the case $p \neq q$, let $g(z) = z/(1 - \frac{1 - e^{-z}}{z})$. Now, using the fact that $g(z) \leq 2 + z$ and the Cauchy-Schwarz inequality, the result follows. □

**Lemma 4.** *For $i \in [n]$ and $p = q$,*

$$\mathrm{Var}[Z_i] \leq 2m_1^2 m_2^2 \Pr[X_i + Y_i > 0], \text{ and hence } \mathrm{Var}[Z_A] = O(m_1^3 m_2^2).$$

*For $p_i \geq q_i$, $\mathrm{Var}[Z_i] \leq O(m_1^3 m_2^2 p_i)$, and for $p_i < q_i$*

$$\mathrm{Var}[Z_i] \leq O(m_1^3 m_2^2) \min\left\{\frac{q_i^2}{p_i}, m_1 q_i^2\right\}. \quad (17)$$

*Proof.* The variance of $Z_i$ can be computed by using the formula for conditional variance. Define,

$$G_i(\sigma) := \mathrm{Var}[(m_2 X_i - m_1 Y_i)^2 - (m_2^2 X_i + m_1^2 Y_i)|X_i + Y_i = \sigma].$$

Let $\beta_i = \frac{m_1 p_i}{m_1 p_i + m_2 q_i}$. Using formulas for binomial moments the conditional variance

$$G_i(\sigma) = F_i(\sigma) + L_i(\sigma),$$

where

$$F_i(\sigma) = 2\beta_i^2 (1 - \beta_i)^2 \sigma(\sigma - 1)(m_1 + m_2)^4,$$

and

$$L_i(\sigma) = 4\beta_i(1 - \beta_i)\sigma(\sigma - 1)^2 (m_1 + m_2)^4 \left(\frac{m_1}{m_1 + m_2} - \beta_i\right)^2.$$

For $p_i = q_i$, $\beta_i = \frac{m_1}{m_1 + m_2}$ and $L_i(\sigma) = 0$. Also, from the proof of Lemma 4 it can be seen that $\mathrm{Var}[\mathbb{E}[Z_i|X_i + Y_i = \sigma]] = 0$, when $p_i = q_i$. Therefore, for $p_i = q_i$,

$$\mathrm{Var}[Z_i] = \mathbb{E}[G_i(\sigma)/\sigma^2] = \mathbb{E}[F_i(\sigma)/\sigma^2] \leq 2m_1^2 m_2^2 \Pr[X_i + Y_i > 0].$$

Let $z_i = m_1 p_i + m_2 q_i$. Then $\Pr[X_i + Y_i > 0] = 1 - e^{-z_i} \leq z_i$, and $\mathrm{Var}[Z_A] = \sum_{i \in A} \mathrm{Var}[Z_i] = O(m_1^3 m_2^2)$.

To prove the bound in the case $p_i \neq q_i$, note that $F_i(\sigma) = 0$, for $\sigma = 0, 1$ and $F_i(\sigma) \leq 2\beta_i^2(1 - \beta_i)^2 \sigma^2 (m_1 + m_2)^4$, for $\sigma \geq 2$. Therefore,

$$
\begin{aligned}
\mathbb{E}\left(\frac{F_i(\sigma)}{\sigma^2}\right) &\leq 2(m_1 + m_2)^4 \beta_i^2 (1 - \beta_i)^2 \Pr[\sigma \geq 2] \\
&\leq 2(m_1 m_2)^2 (m_1 + m_2)^4 \left\{ \frac{p_i^2 q_i^2 (1 - e^{-z_i} - z_i e^{-z_i})}{z_i^4} \right\} \\
&\leq O(m_1^6 m_2^2) \left\{ \frac{p_i^2 q_i^2 \min\{z_i, z_i^2\}}{z_i^4} \right\}.
\end{aligned} \tag{18}
$$

Now, for $p_i \geq q_i$, $z_i \geq \frac{m_1 + m_2}{2}(p_i + q_i)$, and

$$
\mathbb{E}\left(\frac{F_i(\sigma)}{\sigma^2}\right) \leq O(m_1^6 m_2^2) \left\{ \frac{p_i^2 q_i^2 \min\{1, z_i\}}{z_i^3} \right\} \leq O(m_1^3 m_2^2) \left\{ \frac{p_i^2 q_i^2}{(p_i + q_i)^3} \right\} \leq O(m_1^3 m_2^2 p_i).
$$

The remaining terms in the variance can be bounded similarly, and for $p_i \geq q_i$, it follows that $\mathrm{Var}[Z_i] \leq O(m_1^3 m_2^2 p_i)$.

For the case $p_i < q_i$, use the bound $z_i \geq m_1 p_i$ in (18) to get

$$
\mathbb{E}\left[\frac{F_i(\sigma)}{\sigma^2}\right] \leq O(m_1^3 m_2^2) \min\left\{ \frac{q_i^2}{p_i}, m_1 q_i^2 \right\}. \tag{19}
$$

Similarly, $L_i(\sigma) = 0$ for $\sigma = 0, 1$ and $L_i(\sigma) \leq 4\beta_i(1 - \beta_i)\sigma^3(m_1 + m_2)^4 \left(\frac{m_1}{m_1 + m_2} - \beta_i\right)^2$. Therefore, for the case $p_i < q_i$, using the bound $z_i^3 \geq m_1^2 m_2 p_i^2 q_i$, for $z_i \leq 1$, and $z_i^2 \geq m_1 m_2 p_i q_i$, for $z_i \geq 1$ we get

$$
\begin{aligned}
\mathbb{E}\left(\frac{L_i(\sigma)}{\sigma^2}\right) &\leq 4(m_1 + m_2)^4 \beta_i(1 - \beta_i)\left(\frac{m_1}{m_1 + m_2} - \beta_i\right)^2 \mathbb{E}[\sigma \mathbf{1}\{\sigma \geq 2\}] \\
&= 4m_1^3 m_2^3 (m_1 + m_2)^2 \frac{p_i q_i (p_i - q_i)^2 z_i (1 - e^{-z_i})}{z_i^4} \\
&\leq O(m_1^5 m_2^3) \frac{p_i q_i (p_i - q_i)^2 \min\{1, z_i\}}{z_i^3} \\
&= O(m_1^3 m_2^2) \min\left\{ \frac{q_i^2}{p_i}, m_1 q_i^2 \right\}.
\end{aligned} \tag{20}
$$

Finally, from Lemma 3 when $p_i < q_i$

$$
\begin{aligned}
\mathrm{Var}[\mathbb{E}[Z_i | X_i + Y_i = \sigma]] &= (m_1 + m_2)^2 \mathrm{Var}[\sigma]\left(\frac{m_1}{m_1 + m_2} - \beta_i\right)^2 \\
&= m_1^4 m_2^4 \frac{(q_i - p_i)^4}{z_i^3} \\
&\leq O(m_1^3 m_2^2) \min\left\{ \frac{q_i^2}{p_i}, m_1 q_i^2 \right\}.
\end{aligned} \tag{21}
$$

Combining (19), (20), and (21), the variance (17) follows. $\qquad\square$

For the analysis of the algorithms we also need bounds on the $s$-th moment of $Z_A$ corresponding to a set $A$ with the property that for all $i \in A$, $p_i \leq 2b'$ and $q_i \leq 2b'$, where $b' = \frac{256 \log n}{m_2}$, as define in Algorithm 3.

**Lemma 5.** *For any $s \in \mathbb{N}$, and set $A \subset [n]$ such that for all $i \in A$, $p_i \leq 2b'$ and $q_i \leq 2b'$,*

$$
\mathbb{E}[|Z_A - \mathbb{E}[Z_A]|^s] \leq \widetilde{O}_s(m_1^{2s} m_2),
$$

*where $\widetilde{O}_s$ suppresses a factor of $\log^{O(s)} n$.*

*Proof.* Trivially, $|Z_i| \le 3m_2^2 X_i + 3m_1^2 Y_i$. Since $\mathbb{E}[X_i^s]$ is a degree $s$ polynomial in $m_1 p_i$, $\mathbb{E}[X_i^s] = O_s(\max\{m_1^s p_i^s, m_1 p_i\})$. Similarly, for $\mathbb{E}[Y_i^s] = O_s(\max\{m_2^s q_i^s, m_2 q_i\})$. Therefore, for $i \in A$,

$$\mathbb{E}[|Z_i|^s] = O_s(m_2^{2s}\mathbb{E}[X_i^s] + m_1^{2s}\mathbb{E}[Y_i^s]) \quad = \quad \widetilde{O}_s(m_1^{2s} m_2 \max\{p_i, q_i\}). \tag{22}$$

Similarly, $\mathbb{E}[|Z_i|]^s = \widetilde{O}_s(m_1^{2s} m_2 \max\{p_i, q_i\})$, and

$$\mathbb{E}[|Z_A - \mathbb{E}[Z_A]|^s] \le O_s \left( \sum_{i \in A} \mathbb{E}[|Z_i|^s] + \mathbb{E}[|Z_i|]^s \right) \le \widetilde{O}_s(m_1^{2s} m_2). \tag{23}$$

Combining (22) and (27) the lemma follows. $\square$

For the analysis of the algorithm in the extreme case, we will need bounds on the $s$-th moment of $Z_A$ corresponding to a set $A$ with the property that, for all $i \in A$, $\frac{\varepsilon^{2/3}}{20 m_2 n^{1/3}} \le p_i \le 2b'$ and $q_i \le 2b'$. In this case, a more careful analysis gives a better bound on the moments of $Z_A$.

**Lemma 6.** *For any $s \in \mathbb{N}$, and a set $A \subset [n]$ such that for all $i \in A$, $\frac{\varepsilon^{2/3}}{20 m_2 n^{1/3}} \le p_i \le 2b'$ and $q_i \le 2b'$,*

$$\mathbb{E}[|Z_A - \mathbb{E}[Z_A]|^s] \le \tilde{O} \left( \frac{n^{s/3} m_1^s m_2^{s+1}}{\varepsilon^{2s/3}} \right),$$

*where $\widetilde{O}_s$ suppresses a factor of $\log^{O(s)} n$.*

*Proof.* From the definition of $Z_i$,

$$|Z_i| \le O \left( \frac{m_2^2 X_i^2 + m_1^2 Y_i^2}{X_i + Y_i} \right).$$

Conditioned on $X_i + Y_i = \sigma$, $X_i \sim \text{Bin}(\sigma, m_1 p_i / z_i)$ and $Y_i \sim \text{Bin}(\sigma, m_2 q_i / z_i)$, where $z_i = m_1 p_i + m_2 q_i$. Then, $\mathbb{E}[X_i] = \sigma m_2 q_i / z_i := x_i$, and for any $s \ge 1$,

$$\mathbb{E}[X_i^s | X_i + Y_i = \sigma] = O(\max\{x_i, x_i^s\}).$$

Similarly,

$$\mathbb{E}[Y_i^s | X_i + Y_i = \sigma] = O(\max\{y_i, y_i^s\}) \text{ where } \mathbb{E}[Y_i] = \sigma m_2 q_i / z_i := y_i.$$

Therefore, for $\sigma > 0$,

$$\begin{aligned}
\mathbb{E}[|Z_i|^s | X_i + Y_i = \sigma] &\le O_s \left( \max \left\{ \frac{m_1^{2s} m_2^{2s} q_i^{2s} \sigma^s}{z_i^{2s}}, \frac{m_1^{2s} m_2 q_i}{\sigma^{s-1} z_i^{s+1}} \right\} \right) \\
&\le O_s \left( \max \left\{ \frac{m_1^{2s} m_2^{2s} q_i^{2s} \sigma^s}{z_i^{2s}}, \frac{m_1^{2s} m_2 q_i}{z_i^{s+1}} \right\} \right). \tag{24}
\end{aligned}$$

Note that $\mathbb{E}[\sigma] = z_i$ and $\mathbb{E}[\sigma^s] = O_s(z_i^s)$ because $z_i \ge 1$ by assumption. Using $q_i \le 2b'$ we get

$$O_s \left( \frac{m_1^{2s} m_2^{2s} q_i^{2s}}{z_i^s} \right) \le O_s \left( \frac{m_1^s m_2^{2s} q_i^{2s}}{p_i^s} \right) \le O_s \left( \frac{m_1^s m_2^{2s} b'^{2s-1} q_i}{p_i^s} \right) = \tilde{O}_s \left( \frac{m_1^s m_2 q_i}{p_i^s} \right). \tag{25}$$

Moreover, because $m_1 p_i \ge 1$,

$$O_s \left( \frac{m_1^{2s} m_2 q_i}{z_i^{s+1}} \right) \le O_s \left( \frac{m_1^{s-1} m_2 q_i}{p_i^{s+1}} \right) \le O_s \left( \frac{m_1^s m_2 q_i}{p_i^s} \right). \tag{26}$$

Combining (25) and (26) with (24) and using $p_i \ge \frac{\varepsilon^{2/3}}{20 m_2 n^{1/3}}$ (since $i \in A$) gives

$$\mathbb{E}[|Z_i|^s] \le \tilde{O}_s \left( \frac{m_1^s m_2 q_i}{p_i^s} \right) \le \tilde{O}_s \left( \frac{n^{s/3} m_1^s m_2^{s+1} q_i}{\varepsilon^{2s/3}} \right).$$

Similarly, it can be shown that $\mathbb{E}[|Z_i|]^s = \tilde{O}_s \left( \frac{n^{s/3} m_1^s m_2^{s+1} q_i}{\varepsilon^{2s/3}} \right)$, and

$$\mathbb{E}[|Z_A - \mathbb{E}[Z_A]|^s] \le O_s \left( \sum_{i \in A} \mathbb{E}[|Z_i|^s] + \mathbb{E}[|Z_i|]^s \right) \le \widetilde{O} \left( \frac{n^{s/3} m_1^s m_2^{s+1}}{\varepsilon^{2s/3}} \right). \tag{27}$$

completing the proof of the lemma. $\square$

# C  Proof of Proposition 1

We begin by establishing that, with high probability over the first set of samples, $S_1, T_1$, the sets $B, M, H$ successfully partition the elements in the "heavy", "medium", and "light" sets. The proof follows from a union bound over Poisson tail bounds.

**Definition 3.** *Let $b, b'$ be as defined in Algorithm 3. The set $B$ is said to be* faithful *if for all $i \in B$, $p_i > b/2$ or $q_i > b/2$. Similarly, $M$ is said to be* faithful *if for all $i \in M$, $b'/2 \leq \max\{p_i, q_i\} \leq 2b$. Finally, $H$ is said to be* faithful *if $p_i < 2b'$ and $q_i < 2b'$, for all $i \in H$.*

**Lemma 7.** *With probability at least $1 - o(1/n)$ over the randomness in the samples $S_1, T_1$, the sets $B, M,$ and $H$ will be "faithful".*

*Proof.* We leverage the following Chernoff style bound for Poisson distributions: for any $\lambda \leq c$, and $\delta \in (0, 1)$,

$$\Pr\left[|\operatorname{Pois}(\lambda) - \lambda| > \delta c\right] \leq 2e^{-\delta^2 c/3}.$$

Let $X_i^{S_1}$ denote the number of occurrences of $i$ in the $\operatorname{Pois}(m_1)$ samples, $S_1$, drawn from $p$, and $Y_i^{T_1}$ denote the number of occurrences of $i$ in the $\operatorname{Pois}(m_2)$ samples from $q$ that comprise $T_1$. For any domain element $i$ with probability $p_i \geq b'/2$,

$$\Pr\left[|X_i^{S_1} - m_1 p_i| \geq \frac{1}{2} m_1 p_i\right] \leq 2e^{-\frac{1}{4\cdot3} m_1 p_i} \leq 2e^{-20 \log n} = o(1/n^2).$$

Similarly, for any domain element $i$ with probability $q_i \geq b'/2$,

$$\Pr\left[|Y_i^{T_1} - m_2 q_i| \geq \frac{1}{2} m_2 q_i\right] \leq 2e^{-\frac{1}{4\cdot3} m_2 q_i} \leq 2e^{-20 \log n} = o(1/n^2).$$

So far, this ensures that common elements do not occur too infrequently. To ensure that none of the rare elements occur too frequently, note that the same bound implies that for any domain element $i$ with probability $p_i \leq b'/2$,

$$\Pr\left[X_i^{S_1} \geq b' m_1\right] \leq \Pr\left[|X_i^{S_1} - m_1 p_i| \geq b' m_1/2\right] \leq 2e^{-b' m_1/6} \leq 2e^{-20 \log n} = o(1/n^2).$$

Analogously for any domain element $i$ with probability $q_i \leq b'/2$,

$$\Pr\left[Y_i^{T_1} \geq b' m_2\right] \leq \Pr\left[|Y_i^{S_1} - m_2 q_i| \geq b' m_2/2\right] \leq 2e^{-b' m_2/6} \leq 2e^{-20 \log n} = o(1/n^2).$$

Note that if, for all domain elements $i$ with $p_i \geq b'/2$, $|X_i^{S_1} - m_1 p_i| < \frac{1}{2} m_1 p_i$, and for all elements $i$ with $p_i \leq b'/2$, $X_i^{S_1} \leq b' m_1$, and the analogous statements hold for $q_i$ and $Y_i^{T_1}$, then the sets $B, M,$ and $H$ will all be "faithful". By our above bounds, and a union bound over the $n$ elements, with probability at least $1 - o(1/n)$ this occurs. □

We now prove the correctness of Algorithm (3) by establishing that in the case that $p = q$, the algorithm will output ACCEPT with probability at least $2/3$, and in the case that $||p - q||_1 \geq \varepsilon$ the algorithm will output REJECT with probability at least $2/3$. The analysis of these two cases is split into Lemmas 8 and 12. Together with Lemma 7, this establishes Proposition 1:

## C.1  $||p - q||_1 = 0$

We analyze the statistics of the algorithm in the case $p = q$, with respect to the randomness in the samples $S_2, T_2$ under the assumption that the sets $B, M, H$ are faithful.

**Lemma 8.** *Given that the sets $B, M,$ and $H$ are "faithful" and that $p = q$, then with high probability over the randomness in $S_2, T_2$, Algorithm 3 will output ACCEPT.*

*Proof.* **C.1.1 The statistic $V_B$:**

By Lemma 1,

$$\mathbb{E}[V_B] \leq \left(\frac{2|B|}{m_2}\right)^{1/2} + \sum_{i \in B} |p_i - q_i| = \left(\frac{2|B|}{m_2}\right)^{1/2}.$$

From our definition of "faithful", every element of $i \in B$ must have $p_i + q_i \geq b/2 = \frac{128 \log n}{\varepsilon^2 m_2}$, hence $|B| \leq \frac{2\varepsilon^2 m_2}{128 \log n} < \frac{\varepsilon^2 m_2}{64 \log n}$, and

$$\mathbb{E}[V_B] \leq \left(\frac{2|B|}{m_2}\right)^{1/2} \leq \varepsilon \frac{\sqrt{2}}{8\sqrt{\log n}} < \varepsilon/8, \text{ for } n > 2.$$

From Lemma 1, $\text{Var}[V_B] \leq \frac{1}{m_1} + \frac{1}{m_2} \leq \frac{\varepsilon^2}{\sqrt{n}} = o(\varepsilon^2)$. Hence, by Chebyshev's inequality, $\Pr[V_B > \varepsilon/6] \leq o(1)$, and the first check of Algorithm 3 will pass.

**C.1.2 The statistic $W_M$:**

From Lemma 2, $\mathbb{E}[W_M] = m_1^2 m_2^2 \sum_{i \in M} (p_i - q_i)^2 = 0$. Additionally,

$$\text{Var}[W_M] = 2m_1^2 m_2^2 \sum_{i \in M} (m_2 p_i + m_1 q_i)^2 \leq 2m_1^2 m_2^2 \cdot \max_i \{m_2 p_i + m_1 q_i\} \sum_i (m_2 p_i + m_1 q_i).$$

From the fact that $M$ is faithful, $\max_i \{m_2 p_i + m_1 q_i\} \leq O(\frac{m_1 \log n}{m_2 \varepsilon^2})$, and hence we conclude that $\text{Var}[W_M] = O(\frac{m_1^4 m_2 \log n}{\varepsilon^2})$.

By Chebyshev's inequality, and the assumption that $\varepsilon > 1/n^{1/12}$,

$$\Pr\left[W_M \geq \frac{\varepsilon^2 m_1^2 m_2 \log n}{2}\right] = o(1),$$

and hence the second check of Algorithm 3 will pass.

**C.1.3 The statistic $Z_H$:**

By Lemma 3, $\mathbb{E}[Z_H] = 0$, and by Lemma 4, $\text{Var}[Z_H] = O(m_1^3 m_2^2)$. Therefore, by Chebyshev's inequality $\Pr[Z_H \geq C_\gamma m_1^{3/2} m_2] \leq O(\frac{1}{C_\gamma^2})$, which can be made arbitrarily small for a sufficiently large constant $C_\gamma$, and hence the third check of Algorithm 3 will pass. $\square$

## C.2 $||p - q||_1 \geq \varepsilon$

We now consider the execution of the algorithm when $||p - q||_1 \geq \varepsilon$.

**Lemma 9.** *Given that the sets $B, M$, and $H$ are "faithful" and $||p - q||_1 \geq \varepsilon$, then with high probability over the randomness in $S_2, T_2$, Algorithm 3 will output REJECT.*

*Proof.* The proof proceeds by considering the following three cases, at least one of which holds: 1) $\sum_{i \in B} |p_i - q_i| \geq \varepsilon/3$, 2) $\sum_{i \in M} |p_i - q_i| \geq \varepsilon/3$, and 3) $\sum_{i \in H} |p_i - q_i| \geq \varepsilon/3$.

### C.2.1 $\sum_{i \in B} |p_i - q_i| \geq \varepsilon/3$

By Lemma 1, $\mathbb{E}[V_B] \geq \sum_{i \in B} |p_i - q_i| \geq \varepsilon/3$ and $\text{Var}[V_B] \leq \frac{1}{m_1} + \frac{1}{m_2} \leq 2/\sqrt{n}$ Therefore by Chebyshev's inequality, $\Pr[V_B < \varepsilon/6] = o(1)$, and hence the algorithm will output REJECT with high probability.

## C.2.2 $\sum_{i\in M} |p_i - q_i| \geq \varepsilon/3$

From Lemma 2, $\mathbb{E}[W_M] = m_1^2 m_2^2 \sum_{i\in M}(p_i - q_i)^2$. From the definition of "faithful", it follows that $|M| \leq 2\left(\frac{m_2}{128 \log n}\right)$, and hence by Cauchy-Schwarz,

$$(m_1^2 m_2^2)\sum_{i\in M}(p_i - q_i)^2 \geq (m_1^2 m_2^2)\frac{\left(\sum_{i\in M}|p_i - q_i|\right)^2}{|M|} \geq (m_1^2 m_2^2)\frac{128\,\varepsilon^2 \log n}{18 m_2} \geq 7\,\varepsilon^2\, m_1^2 m_2 \log n.$$

Furthermore, from Lemma 2,

$$\mathrm{Var}[W_M] \leq 2 m_1^2 m_2^2 \sum_{i\in M} z_i^2 + 4 m_1^3 m_2^3 \sum_{i\in M} z_i(p_i - q_i)^2,$$

where $z_i = m_1 q_i + m_2 p_i$. As in the proof of Lemma 8, the first term is $O(\frac{m_1^4 m_2 \log n}{\varepsilon^2})$. For the second term, noting that $\sum_i z_i \leq m_1 + m_2$, and $(p_i - q_i)^2 \leq O(\frac{\log^2 n}{\varepsilon^4 m_2^2})$, we get the bound of $O(\frac{m_1^4 m_2 \log n}{\varepsilon^4})$.

By Chebyshev's inequality and the assumption that $\varepsilon > 1/n^{1/12}$, with probability $1 - o(1)$, $W_M > \varepsilon^2\, m_1^2 m_2 \log n$, and the algorithm will output REJECT.

## C.2.3 $\sum_{i\in H} |p_i - q_i| \geq \varepsilon/3$

From Lemma 3, $\mathbb{E}[Z_H] \geq \Omega(\frac{m_1^2 m_2^2 \varepsilon^2}{n})$. Using the assumption that $m_2 = \Omega(\frac{n}{\varepsilon^2 \sqrt{m_1}})$, we conclude that

$$\mathbb{E}[Z_H] = \Omega(m_1^{3/2} m_2).$$

Using the moment bounds from Lemma 5 and the definition of "faithful", for any integer $s > 0$, $\mathbb{E}[|Z_H - \mathbb{E}[Z_H]|^s] \leq \tilde{O}_s(m_1^{2s} m_2)$. By Markov's inequality,

$$
\begin{aligned}
\Pr[Z_H \leq C_\gamma m_1^{3/2} m_2] &\leq \Pr\left[|Z_H - \mathbb{E}[Z_H]| \geq \Omega(m_1^{3/2} m_2)\right] \\
&= \Pr\left[|Z_H - \mathbb{E}[Z_H]|^s \geq \Omega(m_1^{3s/2} m_2^s)\right] \\
&\leq \tilde{O}_s\left(\frac{m_1^{2s} m_2}{m_1^{3s/2} m_2^s}\right) = \tilde{O}_s\left(\frac{m_1^{\frac{s}{2}}}{m_2^{s-1}}\right).
\end{aligned}
$$

As long as $\frac{m_1}{m_2^2} \leq 1/n^c$ for some positive constant $c$, there will be some integer $s_c$, dependent on $c$ for which this probability is $o(1)$. Note that the stipulation in the proposition statement, that $m_1 = O\left((n/\varepsilon^2)^{1-\gamma}\right)$, for some constant $\gamma > 0$, ensures that $\frac{m_1}{m_2^2} = O(1/n^{-2\gamma})$, and hence the algorithm will output REJECT with probability $1 - o(1)$ in this case. $\qquad\square$

# D  Proof of Proposition 2

In this section we prove Proposition 2, showing that Algorithm 4 performs as claimed in the *extreme case* where $m_1 \approx n$. The algorithm is a slight modification of Algorithm (3), tailored to handle the imbalance between the sample sizes from $p$ and $q$. We prove that this algorithm works whenever $m_1 = \Omega\left((n/\varepsilon^2)^{8/9+\gamma}\right)$ for some $\gamma > 0$, and overlaps with the regime of parameters for which the non-extreme algorithm, Algorithm 3, will succeed.

We begin the proof of the above proposition by considering the statistic $R_H$.

**Observation 1.** *Define $R_A = \sum_{i\in A} \frac{\mathbf{1}\{Y_i = 2\}}{X_i + 1}$, for $A \subseteq [n]$. Then*

$$\mathbb{E}[R_A] = \sum_{i=1}^{n} \frac{m_2^2 q_i^2\left(1 - e^{-m_1 p_i}\right)e^{-m_2 q_i}}{2 m_1 p_i}. \tag{28}$$

*Proof.* Since $X_i \sim \text{Pois}(m_1 p_i)$, $\mathbb{E}[\frac{1}{X_i+1}] = \frac{1-e^{-m_1 p_i}}{m_1 p_i}$. Also, $Y_i \sim \text{Pois}(m_2 q_i)$ implies $\Pr[Y_i = 2] = \frac{(m_2 q_i)^2}{2} e^{-m_2 q_i}$. The expectation of $R_A$ now follows from linearity of expectation and the independence of $X_i$ and $Y_i$. $\qquad\square$

As mentioned before, in the extreme case the statistic $Z_A$ can incur a variance of $O(n^4)$, which is at the threshold of what can be tolerated. The statistic $R_A$ is tailored to deal with these cases. This is formalized in the following lemmas: whenever the variance of $Z_A$ is at least the tolerance threshold $\Omega(m_1^3 m_2^2)$, the expected values of $R_A$ in the case $p = q$ is well separated from the likely values of $R_A$ in case $||p - q||_1 > \varepsilon$.

**Lemma 10.** *If $p = q$, $\mathbb{E}[R_A] \leq \frac{m_2^2}{2m_1}$. If $p \neq q$ and $\max_{i \in A} q_i \leq \frac{10}{m_2}$ and $\text{Var}[Z_A] = \Omega(m_1^3 m_2^2)$, then $\mathbb{E}[R_A] \geq \Omega(m_2^2/m_1)$.*

*Proof.* If $p = q$, then

$$\mathbb{E}[R_A] = \frac{m_2^2}{2m_1} \sum_{i \in A} \frac{q_i^2 \left(1 - e^{-m_1 p_i}\right) e^{-m_2 q_i}}{2p_i} \leq \frac{m_2^2}{2m_1} \sum_{i \in A} \frac{q_i^2}{2p_i} \leq \frac{m_2^2}{2m_1}.$$

Now, suppose $p \neq q$. Let

$$A_0 := \{i \in A : m_1 p_i \geq 1/2\}.$$

Note that $\text{Var}[Z_A] \geq \Omega(m_1^3 m_2^2)$ implies that either $\sum_{i \in A_0} \frac{q_i^2}{p_i} \geq C$ or $m_1 \sum_{i \in A \setminus A_0} q_i^2 \geq C$ for some constant $C$ (since by Lemma 4, $\text{Var}[Z_A] \leq O(m_1^3 m_2^2) \sum_{i \in A} \min\left\{ \frac{q_i^2}{p_i}, m_1 q_i^2 \right\}$). We consider the two cases separately:

**1** Suppose $\sum_{i \in A_0} \frac{q_i^2}{p_i} \geq C$. Since $q_i \leq 10/m_2$ for all $i \in A$, it holds that for $i \in A_0, e^{-m_2 q_i} \geq e^{-10}$. Moreover, $i \in A_0$ implies $1 - e^{-m_1 p_i} \geq 1 - e^{-1/2}$. Therefore,

$$\sum_{i \in A_0} \frac{m_2^2 q_i^2 \left(1 - e^{-m_1 p_i}\right) e^{-m_2 q_i}}{2m_1 p_i} \geq \frac{e^{-12} m_2^2}{m_1} \sum_{i \in A_0} \frac{q_i^2}{p_i} \geq \frac{C \cdot e^{-12} m_2^2}{m_1}.$$

**2** Suppose $m_1 \sum_{i \in A \setminus A_0} q_i^2 \geq C$. Using the inequality $1 - e^{-x} \geq x - x^2/2$,

$$
\begin{aligned}
\sum_{i \in A \setminus A_0} \frac{m_2^2 q_i^2 \left(1 - e^{-m_1 p_i}\right) e^{-m_2 q_i}}{2m_1 p_i} &\geq \frac{e^{-10} m_2^2}{2m_1} \sum_{i \in A \setminus A_0} \frac{q_i^2 \left(m_1 p_i - \frac{m_1^2 p_i^2}{2}\right)}{p_i} \\
&= \frac{e^{-10} m_2^2}{2m_1} \sum_{i \in A \setminus A_0} (m_1 q_i^2 - m_1^2 q_i^2 p_i / 2) \\
&\geq \frac{e^{-10} m_2^2}{2} \sum_{i \in A \setminus A_0} (q_i^2 - q_i^2/4) \\
&= \frac{e^{-10} m_2^2}{2} \sum_{i \in A \setminus A_0} 3q_i^2/4 \geq \frac{C \cdot 3e^{-10} m_2^2}{8},
\end{aligned}
$$

where the second to last inequality uses that assumption that $m_1 p_i < 1/2$ for $i \in A \setminus A_0$.

Combining the above cases it follows that $\mathbb{E}[R_A] \geq \Omega(m_2^2/m_1)$. $\qquad\square$

From the proof of the above lemma it is clear that we can choose some absolute constant $K$ such that whenever $p \neq q$ and

$$\max_{i \in A} |q_i| \leq 10/m_2, \quad \text{Var}[Z_A] \geq K m_1^3 m_2^2, \tag{29}$$

then $\mathbb{E}[R_A] \geq 11m_2^2/2m_1$. Hereafter, fix this constant $K$.

**D.1**  $p = q$

Suppose, $m_1 = \Omega((n/\varepsilon^2)^{8/9+\gamma})$ for some $\gamma > 0$. We analyze the statistics in Algorithm 4 in the case that $p = q$, with respect to the randomness in the samples $S_2, T_2$ under the assumption that the sets $B, M, H$ are faithful.

**Lemma 11.** *Given that the sets $B$, $M$, and $H$ are "faithful" and that $p = q$, then with high probability over the randomness in $S_2, T_2$, Algorithm 4 will output ACCEPT.*

*Proof.* From calculations identical to those in case C.1.1, C.1.2, it follows that

$$\Pr[V_B \geq \varepsilon/6] \leq \frac{1}{100}, \quad \Pr[W_M \geq \frac{\varepsilon^2 m_1^2 m_2 \log n}{2}] \leq \frac{1}{100}, \quad \Pr[Z_H \geq C_2 m_1^{3/2} m_2] \leq \frac{1}{100},$$

when $p = q$. Therefore, the unknown distributions will pass the checks in Algorithm 4 that correspond to the statistics $V_B$, $W_M$, and $Z_H$.

It remains to verify the additional two checks in Algorithm 4.

### D.1.1  Check (1) in Algorithm 4

To show that the first check in Algorithm 4 passes, we will show that when $p = q$,

$$\Pr\left[ \text{ there exists } i \in [n] \text{ such that } Y_i \geq 3 \text{ and } X_i \leq \frac{m_1 \varepsilon^{2/3}}{10 m_2 n^{1/3}} \right] < 1/50.$$

Denote $\lambda = \frac{m_1 \varepsilon^{2/3}}{10 m_2 n^{1/3}} = \Omega\left( \frac{m_1^{3/2} \varepsilon^{8/3}}{n^{4/3}} \right) = \Omega(n^\gamma)$, for some constant $\gamma > 0$, since by assumption, $m_1 = \Omega((n/\varepsilon^2)^{8/9+\gamma})$ for some $\gamma > 0$.

If $p_i > \frac{2\lambda}{m_1}$. Then $\Pr[X_i \leq \lambda] \leq \Pr[\text{Pois}(2\lambda) \leq \lambda] = o(1/n^2)$. On the other hand, if $p_i = q_i \leq \frac{2\lambda}{m_1}$, then

$$\Pr[Y_i \geq 3] \leq \Pr\left[ \text{Pois}\left( \frac{2\lambda m_2}{m_1} \right) \geq 3 \right] = \Pr\left[ \text{Pois}\left( \frac{2\varepsilon^{2/3}}{10 n^{1/3}} \right) \geq 3 \right] < \frac{1}{100n}.$$

Hence by a union bound over all $i \in [n]$, check (1) in Algorithm 4 passes.

### D.1.2  The statistic $R$

Recall that $H = [n] \backslash (B \cup M)$, where $B$ and $M$ are defined in (3). Note that by Lemma 10, when $p = q$,

$$\mathbb{E}[R_H] \leq \frac{m_2^2}{2m_1}.$$

By assumption, $m_2^2/m_1 \geq 1$ and the second criteria for Algorithm 4 rejecting is $R_H > C m_2^2/m_1$, for a large constant $C$. Since $R_H$ is a sum of independent random variables, each of which is in the range $(0, 1)$, a standard Chernoff bound applies, yielding that the probability the algorithm rejects due to this $R_H$ is at most $1/100$. $\qquad\square$

**D.2**  $||p - q||_1 \geq \varepsilon$

**Lemma 12.** *Given that the sets $B$, $M$, and $H$ are "faithful" and that $||p - q||_1 \geq \varepsilon$, then with high probability over the randomness in $S_2, T_2$, Algorithm 3 will output REJECT.*

*Proof.* The proof proceeds by considering the following three cases, at least one of which holds: 1) $\sum_{i \in B} |p_i - q_i| \geq \varepsilon/3$, 2) $\sum_{i \in M} |p_i - q_i| \geq \varepsilon/3$, and 3) $\sum_{i \in H} |p_i - q_i| \geq \varepsilon/3$. Now, if either $\sum_{i \in B} |p_i - q_i| \geq \varepsilon/3$ or $\sum_{i \in M} |p_i - q_i| \geq \varepsilon/3$, then from calculations identical to those in Sections C.2.1, C.2.2 it follows that the algorithm will output REJECT.

Therefore, assume that $\sum_{i \in H} |p_i - q_i| \geq \varepsilon/3$. We begin the proof with the following observation:

**Observation 2.** *Suppose there exists $j \in [n]$ such that $q_j \geq \frac{10}{m_2}$ and $p_j \leq \frac{\varepsilon^{2/3}}{20m_2n^{1/3}}$, then*

$$\Pr\left[\exists i \in [n] s.t. Y_i \geq 3 \text{ and } X_i \leq \frac{m_1\varepsilon^{2/3}}{10m_2n^{1/3}}\right] \geq \frac{9}{10}, \tag{30}$$

*that is, Algorithm 4 fails the first check and REJECTS.*

*Proof.* Given $j$ with $q_j \geq \frac{10}{m_2}$ and $p_j \leq \frac{\varepsilon^{2/3}}{20m_2n^{1/3}}$, $\Pr[Y_j \geq 3] > 0.99$, and $\Pr\left[X_j < \frac{m_1\varepsilon^{2/3}}{10m_2n^{1/3}}\right] > 1 - o(1)$. □

Given this observation, we may continue under the assumption that for all $i \in [n]$ such that $q_i \geq \frac{10}{m_2}$, $p_i \geq \frac{\varepsilon^{2/3}}{20m_2n^{1/3}}$. Now, define

$$S_0 := \{i \in [n] : q_i \leq 10/m_2\},$$

and consider the following cases:

*Case 1* $\sum_{i \in S_0} |p_i - q_i| \geq \varepsilon/6$. To begin with suppose that $\mathrm{Var}[Z_{S_0}] \leq Km_1^3m_2^2$, with $K$ as defined in (29). Then by Chebyshev's inequality $\Pr[Z_H \leq C_2m_1^{3/2}m_2] \leq \frac{1}{20}$ (since $\mathbb{E}[Z_{S_0}] \geq \Omega(m_1^{3/2}m_2)$ by Lemma 3). Otherwise, $\mathrm{Var}[Z_{S_0}] \geq Km_1^3m_2^2$, in which case, by Lemma 10, $\mathbb{E}[R_{S_0}] \geq \frac{11m_2^2}{2m_1}$; since $R_H \geq R_{S_0}$ is a sum of independent random variables, with values between 0 and 1, a Chernoff bound yields that with probability at least 0.99, $R_H$ will exceed the threshold and the second check of Algorithm 4 will fail.

*Case 2* Finally, suppose that $\sum_{i \in H \setminus S_0} |p_i - q_i| \geq \varepsilon/6$. Since $q_i > 10/m_2$ for all $i \in H \setminus S_0$, it suffices to assume that $p_i \geq \frac{\varepsilon^{2/3}}{20m_2n^{1/3}}$ by Observation 2. From Lemma 6, letting $T = H \setminus S_0$, we have that $\mathbb{E}[Z_T] \geq O(\varepsilon^2 m_1^2m_2^2/36n)$, and

$$\mathbb{E}[|Z_T - \mathbb{E}[Z_T]|^s] = O\left(\frac{n^{s/3}m_1^sm_2^{s+1}}{\varepsilon^{2s/3}}\right).$$

By Markov's inequality,

$$\begin{aligned}
\Pr[Z_T \leq C_\gamma m_1^{3/2}m_2/2] &\leq \Pr[|Z_T - \mathbb{E}[Z_T]| \geq \Omega(m_1^{3/2}m_2)] \\
&\leq \widetilde{O}_s\left(\frac{n^{s/3}m_1^sm_2^{s+1}}{\varepsilon^{2s/3}m_1^{3s/2}m_2^s}\right) \\
&\leq \widetilde{O}_s\left(\frac{n^{s/3}m_2}{\varepsilon^{2s/3}m_1^{s/2}}\right). 
\end{aligned} \tag{31}$$

If $m_2 = \frac{n}{\sqrt{m_1}\varepsilon^2}$ then (31) becomes $\widetilde{O}_s\left(\frac{(n/\varepsilon^2)^{s/3+1}}{m_1^{s/2+1/2}}\right)$. Since $m_2 \geq \Omega((n/\varepsilon^2)^{8/9})$, by taking $s > 5$, we can make the probability in (31) $o(1)$. Similarly, if $m_1 = n$ and $m_2 = \sqrt{n}/\varepsilon^2$, then with $s = 6$, (31) becomes $\widetilde{O}_s\left(\frac{1}{\varepsilon^8\sqrt{n}}\right) = o(1)$ as $\varepsilon \geq n^{-\frac{1}{12}}$. Together with the concentration of $Z_{S_0}$ from Chebyshev's inequality, we get that in this case, the $Z$ statistic check fails and the algorithm will output REJECT with probability at least 0.99 in this case.

□

# E   Lower Bound for $\ell_1$ Testing

In this section, we present lower bounds for the closeness testing problem under the $\ell_1$ norm using the machinery developed in Valiant [29, 30]. To this end, define the $(k_1, k_2)$-based moments $m(r, s)$ of a distribution pair $(p, q)$ as $k_1^r k_2^s \sum_{i=1}^n p_i^r q_i^s$. Valiant [30, Theorem 4.6.9] showed that if the

distributions $p_1^+, p_2^+$ have probabilities at most $1/1000k_1$, and $p_1^-, p_2^-$ have probabilities at most $1/1000k_2$, and

$$\sum_{r+s>1} \frac{|m^+(r,s) - m^-(r,s)|}{\sqrt{1 + \max\{m^+(r,s), m^-(r,s)\}}} < \frac{1}{1000}. \tag{32}$$

then the distribution pair $(p_1^+, p_2^+)$ cannot be distinguished with probability $13/24$ from $(p_1^-, p_2^-)$ by a tester that takes $\mathrm{Pois}(k_1)$ samples from $(p_1^+, p_2^+)$ and $\mathrm{Pois}(k_2)$ samples from $(p_1^-, p_2^-)$.

Using this we prove the following proposition:

**Proposition 3.** *Let $n^{2/3}/\varepsilon^{4/3} \le m_1 \le n$. Then there exists distributions $p$ and $q$ such that given $\Theta(m_1)$ samples from $p$ requires $\Omega(\frac{n}{\sqrt{m_1}\varepsilon^2})$ samples from $q$ to distinguish between $p = q$ and $||p - q||_1 \ge \varepsilon$ with high probability.*

*Proof.* Fix $\delta = 1/4$. Let $b = 1/m_1$ and $a = C/n$, where $C$ is an appropriately chosen constant. Let $A$, $B$, and $C$ be disjoint subsets of size $(1-\delta)/b$, $1/a$, $1/a$, respectively. Consider two distributions

$$p = b\mathbf{1}_A + \delta a \mathbf{1}_B,$$

and

$$q = b\mathbf{1}_A + \delta a(1 + \varepsilon z)\mathbf{1}_B,$$

where $z$ is 1 or -1 depending on whether the index is even or odd (in such a way that $\sum_{i=1}^n q_i = 1$). Then clearly $||p - q||_1 = \delta\varepsilon = \varepsilon/4$.

Define $k_1 = cm_1$ and $k_2 = c\varepsilon^{-2}n/\sqrt{m_1}$, where $c$ is a sufficiently small constant. Then $||p||_\infty = b \le \frac{1}{1000k_1}$ and $||p||_\infty = b \le \frac{1}{1000k_2}$, whenever $m_1 \ge n^{2/3}/\varepsilon^{4/3}$ and $b \ge a$.

Let $(p, p) = (p_1^+, p_2^+)$ and $(p, q) = (p_1^-, p_2^-)$ and computing the $(k_1, k_2)$-based moments gives:

$$m^+(r,s) = k_1^r k_2^s (1-\delta)b^{r+s-1} + k_1^r k_2^s \delta^{r+s} a^{r+s-1},$$

and

$$m^-(r,s) = k_1^r k_2^s (1-\delta)b^{r+s-1} + k_1^r k_2^s \delta^{r+s} a^{r+s-1}\left(\frac{(1+\varepsilon)^s + (1-\varepsilon)^s}{2}\right).$$

By Theorem 4.6.9 of Valiant [30], to show that $(k_1, k_2)$ samples are not enough, it suffices to have (32). Observe,

$$\frac{|m^+(r,s) - m^-(r,s)|}{\sqrt{1 + \max\{m^+(r,s), m^-(r,s)\}}} \le \frac{k_1^r k_2^s \delta^{r+s} a^{r+s-1}\left(1 - \frac{1}{2}((1+\varepsilon)^s + (1-\varepsilon)^s)\right)}{\sqrt{k_1^r k_2^s (1-\delta)b^{r+s-1}}}.$$

For any $s \ge 0$, define $h(\varepsilon, s) = 1 - \frac{(1+\varepsilon)^s + (1-\varepsilon)^s}{2}$ Observe that $h(\varepsilon, 1) = 0$, and $|h(\varepsilon, s)| \le 1$, for $s \ne 1$. Note that $m_1 \ge n^{2/3}/\varepsilon^{4/3}$, implies that $\varepsilon \ge n^{-\frac{1}{4}}$. Therefore, for every fixed $r \ge 0$ and $s \ne 1$,

$$h(\varepsilon, s)k_1^{\frac{r}{2}} k_2^{\frac{s}{2}} b^{-(r+s-1)/2} a^{r+s-1} \le c^{\frac{r+s}{2}}\left(\frac{m_1}{n}\right)^r \left(\frac{m_1^{\frac{1}{2}}}{\varepsilon^2 n}\right)^{\frac{s}{2}-1} \le c^{\frac{r+s}{2}}\left(\frac{m_1}{n}\right)^{r+\frac{s}{4}+\frac{1}{2}} < c^{\frac{r+s}{2}},$$

since $m_1 \le n$ by assumption. This shows (32) if $c$ is chosen small enough. $\square$

The optimality of the $\ell_1$ tester, establishing the lower bound in Theorem 1, follows from the above proposition together with the lower bound of $\sqrt{n}/\varepsilon^2$ for testing uniformity given in Paninski [21].