[Reviews · NeurIPS 2015]

Submitted by Assigned_Reviewer_1

This is a light review and so I have not written detailed comments for the authors. I may change my score and review in response to discussion.

AFTER AUTHOR RESPONSES:

I agree in a large part with Reviewer 3's comments, but I think the responses are reasonable. I think this will be interesting to those at NIPS. I am keeping my scores as-is.
Summary: This is a nice paper leveraging several recent result on distribution and property testing to the problem of testing for unequal sized samples. The really interesting part is on estimating the mixing time of a Markov chain. This is a strong theoretical contribution of interest to that segment of the NIPS community.

Submitted by Assigned_Reviewer_2

To use the test statistics requires knowing the cut-off values which were explicitly derived in the paper. In addition, the authors did not compare experimentally with existing method for equal sample size by using

min(m_1,m_2)

samples or some reasonable modification .

An estimate of the how much benefit in practice do we observe would be nice and necessary to see the practical impact of the proposed scheme.

Summary: The paper proposes a summary statistics to estimate whether two discrete distribution given by samples of unequal size is equal or not.

It is not fully clear what is the importance of such a result in context of existing work. Nevertheless, the problem is interesting and the results seem to advance a very specialized case of unequal sample size.

The authors shows an interesting consequence of estimating the mixing time of Markov chain.

Submitted by Assigned_Reviewer_3

The paper describes a new algorithm to test closeness of two distributions. The algorithm given in Algorithm 1 depends on several "appropriately selected" constants, actually there are four of them. It is not clear how they should be selected and how they might affect the effectiveness of the algorithm. The experiment of synonym identification, however, shows "wolf" and "fox" two different species turns out to be closer than "almost" and "nearly". The author also did not show how traditional z-test, language modeling or more recent word embedding compared to the proposed algorithm. Instead, the authors might want to consult some colleagues in biomedical science where plenty of problems need a more sensitive test like the authors attempt here.
Summary: Though the paper addresses an old problem in statistics but the results are concrete and interesting. However, the description of the algorithm is not clear to me and the results of synonym identification is not convincing.

Submitted by Assigned_Reviewer_4

The paper considers the problem of testing if two samples are from identical or very different distributions. This is an interesting question since in many practical scenarios, one of the data sources might be more precious/hard to obtain than the other.

This paper characterizes the precise tradeoff between the sample lengths required to solve this problem, and also nail down the dependence on the accuracy, epsilon up to constant factors. From my understanding, the actual upper bounds (Achievability) bounds of [3] are actually nearly optimal (up to logarithmic terms), and this paper removes all spurious log-factors and reduces the exponent of epsilon to the optimal value. Even if the upper bound is a small improvement in terms of the number of samples, it still requires new insights and better testers, something the authors have managed to achieve. The lower bounds of the paper are similar to those in [8], [28] and matches the upper bounds shown up to constant factors.

Here are some specific comments on the paper:

In Theorem 1, the authors require eps \ge n^{-1/12}. Is this because there is an additional term with higher powers of

epsilon, which disappears when this happens. If yes, it would be nice if the main result contains the entire complexity, and put the conditional result as a corollary.

98: uniform distribution minimizes the number of domain elements appearing more than once. This is slightly incorrect. Please rephrase as it maximizes the number of elements appearing once.

108-112: I got a little confused with using n for domain size and then for # of samples when describing [1,2]. 217: should m1 be something else. I am not sure if it is n^{1-\gamma}. 270-278: the authors can give some further intuition and requirement of the additional step needed.

321: Pois (O(1/\sqrt n)) ... should it be Pois(O(\sqrt n)).

I am not fully sure what conclusions to draw from the experiments. I might be missing something elementary here.

The paper is well written, and is an enjoyable read.

Summary: The paper studies testing whether two samples of unequal lengths are coming from the same or very different distributions. The upper bounds seem to be incremental (removing log-factors and improving exponents of epsilon), and the lower bounds are strong, and prove optimality. The ideas for the lower bound are similar to those in the literature, but still require some work.

Author Feedback
Author rebuttal: We thank the reviewers for their time and helpful comments. Below we respond to the specific questions of the reviewers.

Reviewer 2: Explicit values of the constants used in the algorithm can be extracted from our proofs, though we chose to state our theoretical result in big-Oh notation to simplify the exposition and proofs rather than chase constants. The purpose of the empirical results was to demonstrate that the statistic at the core of our tests can be fruitfully applied in practice to resolve differences between distributions using surprisingly few samples--even without bothering with constants. For example, traditional tests would require many more samples to determine that 'wolf' and 'fox' have more similar usage than 'almost' and 'nearly'.

Language modeling and such approaches as word embeddings require enormous amounts of data, in part because they work with empirical distributions, and make little effort to account for the sampling errors of the empirical distribution. Our work may suggest that such approaches could benefit from attempts to modify them so as to correct for the effects of sampling.

In regards to finding compelling applications of our algorithms, we are currently working with some bio-engineers who are considering using our tests to compare the distributions of RNA expression between different cells, though that work is more on the practical side and is not really geared towards the NIPS community...

[The 'traditional z-test' does not apply to these settings of testing closeness of distributions, in particular when small samples are drawn from distributions over large alphabets.]

Reviewer 3: The eps > n^(-1/12) condition simplifies the analysis, though can probably be removed. Note that if eps is extremely small (say, eps< n^(-1/2)), then the problem is essentially no longer in the 'large alphabet'/sub-linear sample-size regime, and the techniques become more similar to the classical statistical analysis.

Reviewer 4: We agree that it would be helpful to include a comparison with the performance of using min(m_1,m_2) samples. In the setting of Figure 2, one can do this comparison by contrasting the diagonal to the top or right edges (which is most pronounced in the leftmost plot), though the benefit of using unequal sized samples is more striking in the setting of differences in word usages (Figure 1), though it did not occur to us to include that comparison in the figure.